*Report*

# Transposable element activity captures human pluripotent cell states

Florencia Levin-Ferreyra [ID][1,2,3,4], Srikanth Kodali[1,2,3,4], Yingzhi Cui [ID][1,2,3,4], Alison R S Pashos [ID][5,6,7], Patrizia Pessina [ID][1,2,3,4], Justin Brumbaugh[5,6,7] & Bruno Di Stefano [ID][1,2,3,4] ✉

## Abstract

**Human pluripotent stem cells (hPSCs) exist in multiple, transcriptionally distinct states and serve as powerful models for studying human development. Despite their significance, the molecular determinants and pathways governing these pluripotent states remain incompletely understood. Here, we demonstrate that transposable elements act as sensitive indicators of distinct pluripotent cell states. We engineered hPSCs with fluorescent reporters to capture the temporal expression dynamics of two state-specific transposable elements, LTR5_Hs, and MER51B. This dual reporter system enables real-time monitoring and isolation of stem cells transitioning from naïve to primed pluripotency and further towards differentiation, serving as a more accurate readout of pluripotency states compared to conventional systems. Unexpectedly, we identified a rare, metastable cell population within primed hPSCs, marked by transcripts related to preimplantation embryo development and which is associated with a DNA damage response. Moreover, our system establishes the chromatin factor NSD1 and the RNA-binding protein FUS as potent molecular safeguards of primed pluripotency. Our study introduces a novel system for investigating cellular potency and provides key insights into the regulation of embryonic development.**

**Keywords** Embryonic Stem Cells; Pluripotency; Totipotency; DNA Damage; Transposable Elements
**Subject Categories** DNA Replication, Recombination & Repair; Genetics; Gene Therapy & Genetic Disease; Stem Cells & Regenerative Medicine

## Introduction

Human pluripotent stem cells (hPSCs) are invaluable for studying early embryonic development and exhibit various states of pluripotency (Du and Wu, 2024; Nichols and Smith, 2009; Smith, 2017; Takashima et al, 2014; Theunissen et al, 2014). Each state displays distinct global epigenetic and gene expression profiles, as well as developmental potential (Du and Wu, 2024). For example, naïve hPSCs mirror the developmentally primitive inner cell mass of the preimplantation embryo (Di Stefano et al, 2018; Du and Wu, 2024; Guo et al, 2021; Nichols and Smith, 2009; Weinberger et al, 2016), whereas primed hPSCs resemble the epiblast cells of the post-implantation embryo (Du and Wu, 2024; Nichols and Smith, 2009; Weinberger et al, 2016). Recently, human 8-Cell (8C)-like cells have been isolated in vitro, which, unlike primed and naïve PSCs, express genes involved in zygotic genome activation (ZGA) and possess expanded developmental potential (Mazid et al, 2022; Taubenschmid-Stowers et al, 2022).

Various methods have been developed to modulate embryonic cell potency in vitro, typically involving cocktails of small molecules targeting essential developmental pathways (Ai et al, 2023; Bayerl et al, 2021; Di Stefano et al, 2018; Gafni et al, 2013; Hanna et al, 2010; Khan et al, 2021; Li et al, 2009; Lynch et al, 2020; Mazid et al, 2022; Qin et al, 2016; Takashima et al, 2014; Theunissen et al, 2014; Valamehr et al, 2014; Yu et al, 2022). These protocols can be lengthy, asynchronous, and inefficient, making it difficult to obtain homogeneous populations of embryonic stem cells with defined developmental potential (Collier et al, 2022; Collier et al, 2017; Du and Wu, 2024; Guo et al, 2017). Consequently, the lack of sensitive, specific indicators for human pluripotent cell states has been a significant challenge in optimizing conversion protocols and understanding the mechanisms underlying human developmental cell fate transitions.

In this study, we identify transposable elements as sensitive and accurate indicators of human pluripotent stem cell states. We developed a transposon-based dual reporter system that efficiently tracks the dynamics of cell state transitions from naïve-to-primed pluripotency and further during differentiation. Using this system, we discovered a rare, metastable cell population within primed cells characterized by low expression of genes associated with early preimplantation embryo development and triggered by DNA damage. Finally, we unveiled previously unknown roles for specific chromatin and RNA-binding factors as pluripotency regulators governing the primed-to-naïve conversion.

[1]Stem Cells and Regenerative Medicine Center, Baylor College of Medicine, Houston, TX, USA. [2]Center for Cell and Gene Therapy, Baylor College of Medicine, Houston, TX, USA. [3]Department of Molecular and Cellular Biology, Baylor College of Medicine, Houston, TX, USA. [4]Dan L Duncan Comprehensive Cancer Center, Baylor College of Medicine, Houston, TX, USA. [5]Department of Molecular, Cellular, and Developmental Biology, University of Colorado Boulder, Boulder, CO, USA. [6]University of Colorado Cancer Center, Anschutz Medical Campus, Aurora, CO, USA. [7]Charles C. Gates Center for Regenerative Medicine, University of Colorado Anschutz Medical Campus, Aurora, CO, USA. ✉E-mail: bruno.distefano@bcm.edu

# Results and discussion

## The retroviral elements LTR5_Hs and MER51B are selective indicators of naïve and primed pluripotency

Initially, we sought to define novel molecular characteristics of naïve and primed pluripotency that could be exploited to engineer a reporter system capable of distinguishing PSC states in vitro. Prior studies have suggested that transposable elements (TEs) might provide a sensitive means of differentiating between primed and naïve hPSCs (Di Stefano et al, 2018; Theunissen et al, 2016). Thus, we analyzed TE expression in several human embryonic stem cell (hESC) lines (i.e., UCLA1, -3, -4, -5, and -9) (Di Stefano et al, 2018) cultured under conventional primed conditions as well as naïve conditions (Fig. EV1A and methods). Corroborating previous findings, the transcriptional pattern of TE expression distinguished the primed samples from naïve samples (Buckberry et al, 2023; Di Stefano et al, 2018; Greenberg and Bourc'his, 2019; Guo et al, 2017; Pontis et al, 2019; Theunissen et al, 2016). Further analysis revealed that members of the LTR5_Hs family were primarily induced in the naïve state (Di Stefano et al, 2018; Fuentes et al, 2018; Grow et al, 2015; Theunissen et al, 2016), while the endogenous retrovirus family MER51B, not previously associated with pluripotency, was specifically active in the primed state (Fig. 1A). Analysis of

pluripotency transcription factor occupancy and deposition of histone marks (Figs. EV1B-D) confirmed that LTR5_Hs and MER51B elements are specifically activated in naïve and primed hPSCs, respectively. Additionally, analysis of single-cell transcriptomic (Data Ref: Xue et al, 2013) and chromatin accessibility (Data Ref: Liu et al, 2019) data from early human embryos across several developmental cell states—ranging from oocytes to blastocysts—revealed that LTR5_Hs becomes active starting at the 8-cell stage, while MER51B is transcriptionally silenced during preimplantation development (Fig. EV1E, F). Together, these findings suggest the possibility of utilizing these transposable elements to distinguish between naïve and primed pluripotent stem cell states in vitro.

To test this possibility, we developed a dual fluorescent reporter system based on the repetitive sequences of LTR5_Hs and a MER51B (see methods). The two reporters included a minimal TK promoter, with the fluorescent proteins Turbo-RFP for the LTR5_Hs reporter and a destabilized GFP for MER51B, which were both introduced into UCLA4 hESCs (Fig. 1B). As expected, hESCs cultured under primed conditions uniformly expressed MER51B-GFP but did not exhibit LTR5_Hs-RFP expression, as confirmed by microscopy and flow cytometry (Fig. 1C–E). Conversely, the LTR5_Hs reporter was robustly expressed in naïve hESC cultures (i.e., 5i/LAF (Theunissen et al, 2016; Theunissen et al, 2014) and PXGL (Guo et al, 2021)), which did not express

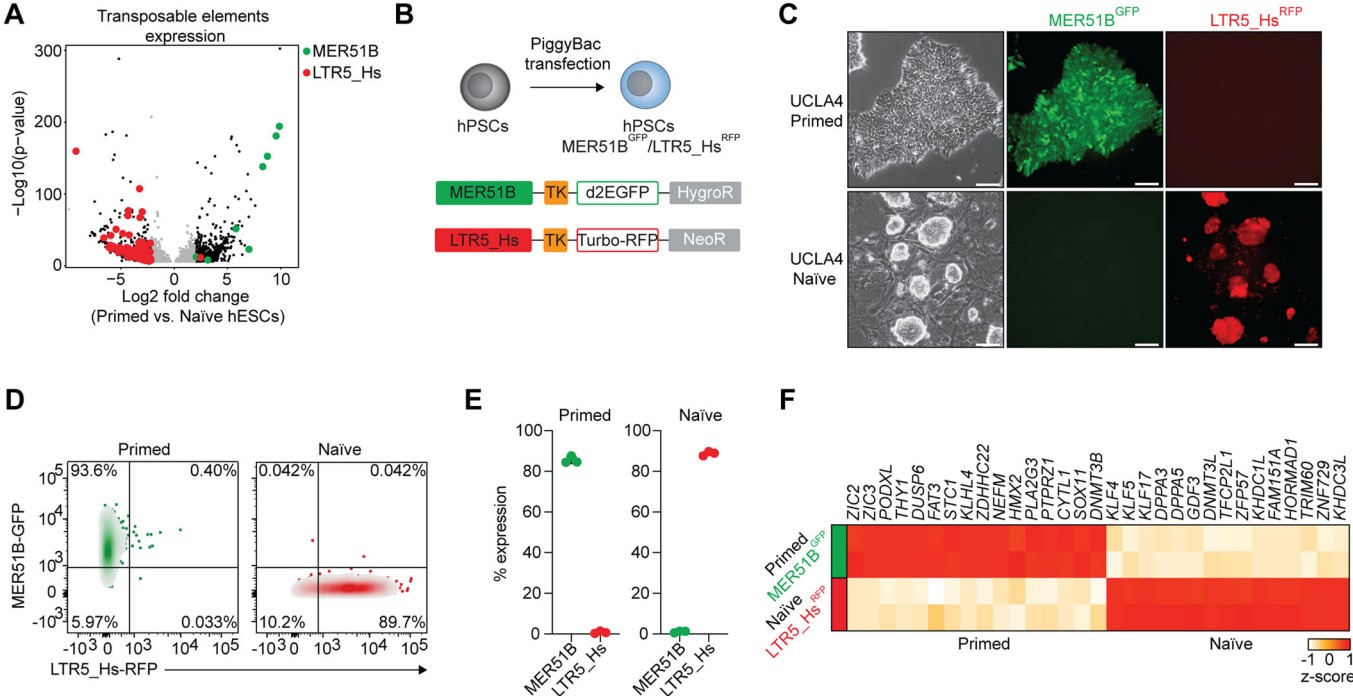

**Figure 1. Differential expression of MER51B and LTR5_Hs distinguishes primed and naïve human pluripotent states.**

(A) Volcano plot of RNA-seq data (Data ref: Di Stefano et al, 2018) showing expression of MER51B (green dots) and LTR5_Hs (red dots) elements in primed ($n = 5$) vs. naïve hESC lines ($n = 14$). A transposable element was considered to be differentially expressed when abs(log$_2$(fold change)) >3 and false discovery rate (FDR) <0.01. Statistical significance was determined by the Wald test with Benjamini–Hochberg correction. (B) Schematic of PiggyBac fluorescent reporter constructs to track MER51B and LTR5_Hs activity. (C) Representative phase contrast and fluorescence images of MER51B-GFP/LTR5_Hs-RFP reporter UCLA4 cells in primed and naïve states. Scale bar: 300 μm. (D) Representative flow cytometry plots showing MER51B-GFP and LTR5_Hs-RFP expression in primed and naïve UCLA4 cells. (E) Flow cytometric quantification of MER51B-GFP and LTR5_Hs-RFP expression in primed and naïve UCLA4 cells. Each data point represents an independent biological replicate ($n = 3$). (F) Heatmap showing expression levels of selected primed and naïve genes in sorted LTR5_Hs-RFP$^+$ naïve or MER51B-GFP$^+$ primed UCLA4 cells. Each column represents an independent biological replicate ($n = 2$ per group). Source data are available online for this figure.

MER51B-GFP (Figs. 1C–E and EV1G). Notably, RNA-seq analysis of sorted GFP$^+$ primed cells and RFP$^+$ naïve hESCs revealed that genes associated with naïve pluripotency (e.g., *KLF17*, *DPPA3*, and *DPPA5*) were highly expressed in LTR5_Hs-RFP$^+$ naïve cells, while genes indicative of primed pluripotency (e.g., *ZIC3*, *THY1*, and *PODXL*) were exclusively expressed in MER51B-GFP$^+$ primed hESCs (Figs. 1F and EV1H).

These results collectively establish LTR5_Hs and MER51B as selective indicators of primed and naïve pluripotent stem cell states.

## LTR5_Hs and MER51B expression dynamics mirror the primed-to-naïve conversion

We subsequently investigated the potential of our dual reporter system to reliably track changes in pluripotent cell state using independent conversion approaches. First, we treated our primed dual reporter cells with 5i/LAF (Theunissen et al, 2014), a cocktail that promotes efficient primed-to-naïve conversion, and monitored the temporal dynamics of MER51B-GFP and LTR5_Hs-RFP activity using flow cytometry (Fig. 2A). We observed a nearly complete loss of MER51B-GFP and significant upregulation of LTR5_Hs-RFP within 48 h, with most cells robustly and exclusively expressing LTR5_Hs-RFP by day 4. Of note, the loss of MER51B and gain of LTR5_Hs expression correlated with the downregulation of the primed-specific surface marker CD90 as well as the induction of the naïve-specific markers CD75, KLF17, DPPA5, and KLF4 (Collier et al, 2017) (Figs. 2B and EV2A,B). Moreover, MER51B expression decreased more rapidly than CD90, suggesting it could serve as a more sensitive indicator of the loss of primed pluripotency (Fig. 2B). Thus, changes in the expression patterns of our transposable element-driven reporters accurately reflect the transition from primed-to-naïve pluripotency.

Next, we used our dual reporter system to monitor the conversion of primed hPSCs by 4CL (Mazid et al, 2022), a culture medium that, in addition to inducing expression of naïve pluripotency genes, activates genes involved in ZGA, a hallmark of the 8-cell (8C) embryo stage (Mazid et al, 2022; Taubenschmid-Stowers et al, 2022) (Fig. 2C). Interestingly, and in contrast to 5i/LAF, 4CL-based conversion of primed dual reporter hPSCs generated a mixed population of cells heterogeneous in their expression of MER51B-GFP and LTR5_Hs-RFP, including a subpopulation that expressed both reporters (Fig. 2C,D). We hypothesized that this heterogeneity might reflect distinct underlying gene expression programs.

To test this hypothesis, we performed RNA-seq from 4CL-induced hPSCs, using 5i/LAF-induced LTR5_Hs-RFP$^+$ hPSCs (naïve) and uninduced MER51B-GFP$^+$ hPSCs (primed) as controls. As expected, 5i/LAF-induced LTR5_Hs-RFP$^+$ hPSCs robustly expressed genes associated with naïve pluripotency, while their uninduced MER51B-GFP$^+$ counterparts exhibited high levels of primed pluripotency-associated genes (Figs. 2E and EV2C). Although a MER51B-GFP$^+$ population persisted under 4CL conversion, its primed pluripotency gene expression program was destabilized, leading to a transcriptional landscape intermediate between naïve and primed hPSCs (Figs. 2E and EV2C). Intriguingly, the MER51B-GFP$^+$/LTR5_Hs-RFP$^+$ population that arose under 4CL conditions exhibited elevated expression of ZGA markers, at levels comparable to those of 8C-like cells expressing

the ZGA-associated marker *TPRX1* (Figs. 2E and EV2D). We corroborated these findings using qRT-PCR, showing that in the 4CL medium, RFP$^+$ and GFP$^+$ cells expressed the highest levels of naïve and primed pluripotency genes, respectively, while GFP$^+$/RFP$^+$ cells expressed preimplantation transcripts (Fig. EV2E). These data suggest that cells undergoing primed-to-naïve conversion have different cell fate trajectories when treated with 5i/LAF or 4CL and suggest that these conversion methods are mechanistically distinct.

We next investigated whether our reporter system could monitor the naïve-to-primed transition in hESCs. We induced re-priming of naïve dual-reporter hESCs by treating them with the Tankyrase inhibitor XAV939 in serum-free medium for 14 days (Rostovskaya et al, 2022; Rostovskaya et al, 2019). By passage 3, we observed a decrease in LTR5_Hs-RFP expression concurrent with the emergence of MER51B-GFP$^+$ cells. Most cells transitioned to a predominantly GFP$^+$/RFP$^-$ state by passage 8 (Fig. 2F). Importantly, these reprimed cells retained the ability to revert to a naïve GFP$^-$/RFP$^+$ state when exposed to 5i/LAF conditions (Fig. EV2F). These results demonstrate that our transposable element-driven reporters faithfully track the naïve-to-primed pluripotency transition, providing a robust system to study dynamic cell state changes in hESCs.

Collectively, we have demonstrated that MER51B and LTR5_Hs can faithfully and reproducibly track the dynamics of hPSC cell state transitions. Moreover, these data highlight a potential use for our dual reporter system to isolate cells undergoing reprogramming to either a naïve state or an 8C-like state.

## MER51B expression tracks exit from pluripotency

To further validate the efficacy of MER51B and LTR5_Hs as selective indicators of pluripotent states, we monitored their activity during exit from pluripotency. We first induced multi-lineage differentiation of primed hPSCs through the formation of embryoid bodies (EBs) and tracked reporter expression over time. EBs facilitate the initiation of spontaneous differentiation and the generation of all three germ layers (i.e., ectoderm, mesoderm, and endoderm). By the fourth day of EB culture, the expression of MER51B-GFP ceased, while key developmental genes, including *T*, *PAX6*, and *SOX7*, became activated (Figs. 2G,H and EV2G,H). In contrast, LTR5_Hs-RFP remained silenced throughout the differentiation process (Fig. 2G, H), confirming its specificity for the naïve state of hPSCs. Notably, the loss of MER51B-GFP expression occurred significantly faster than the downregulation of the pluripotency transcripts *OCT4* and *NANOG* (Fig. 2I). This observation suggests that our reporter system could more effectively monitor the exit from pluripotency, compared to conventional systems based on the expression of pluripotency transcription factors.

To corroborate these findings, we assessed the dynamics of our dual reporter expression upon treatment with the MAPK pathway inhibitor PD0325901 (referred to hereafter as MEKi) (Fig. 2J). This treatment triggers hPSCs to exit pluripotency efficiently and homogeneously (Gonzales et al, 2015). We observed that MER51B-GFP was significantly downregulated by day 2 and became undetectable by day 6 (Fig. 2K). This process correlated with the downregulation of pluripotency genes and the activation of differentiation, as assessed by RNA-seq analysis (Figs. 2L

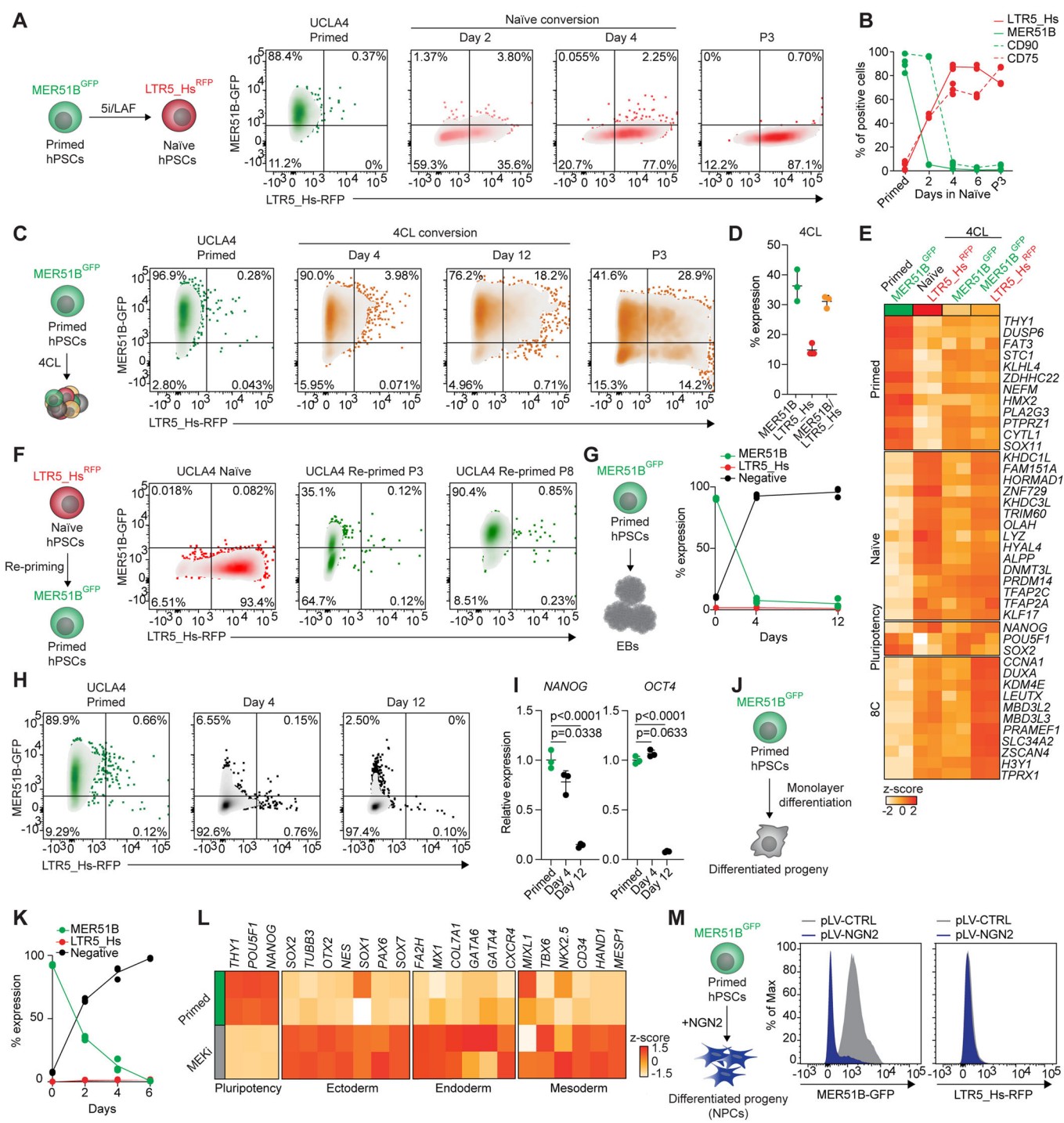

and EV2I). Similar results were observed using an additional hESC line (Fig. EV2J), and an orthogonal approach to induce exit from pluripotency by withdrawing the self-renewal factors bFGF and TGF-β (Gonzales et al, 2015) (Fig. EV2K–M).

Finally, we sought to determine if our reporter could be used to study regulators of pluripotency exit. As proof-of-principle, we overexpressed the transcription factor NGN2 in our dual reporter cell lines and conducted flow cytometric analysis for the expression of MER51B-GFP and LTR5_Hs-RFP. NGN2, a potent regulator of

neurogenesis, induces a rapid and robust exit from pluripotency and differentiation into neural progenitors when overexpressed in primed hPSCs (Lin et al, 2018). As expected, NGN2 overexpression forced hPSCs to exit pluripotency, as indicated by the silencing of the primed-specific MER51B-GFP reporter, without inducing LTR5_Hs-RFP expression (Fig. 2M).

In conclusion, these data establish our transposable element-based reporter system as a robust, tractable tool for monitoring the dissolution of the pluripotent state.

**Figure 2. MER51B and LTR5_Hs activity is dynamically regulated during hPSC state transitions.**

(A) Schematic of 5i/LAF-induced naïve conversion of primed hESCs showing MER51B and LTR5_Hs reporter expression (left panel). Representative flow cytometry plots showing changes in MER51B-GFP and LTR5_Hs-RFP expression during 5i/LAF-induced primed-to-naïve conversion of UCLA4 cells (right panel). (B) Flow cytometric quantification of MER51B-GFP, LTR5_Hs-RFP, CD90, and CD75 expression during primed-to-naïve conversion. Each data point represents an independent biological replicate ($n = 3$). (C) Schematic of 4CL-induced conversion of primed hESCs showing MER51B and LTR5_Hs reporter expression (left panel). Representative flow cytometry plots showing changes in MER51B-GFP and LTR5_Hs-RFP expression during 4CL-induced conversion of UCLA4 cells (right panel). (D) Flow cytometric quantification of the percentage of MER51B-GFP$^+$, LTR5_Hs-RFP$^+$, and MER51B-GFP$^+$LTR5_Hs-RFP$^+$-expressing cells in established 4CL cell cultures (after P3). Each data point represents an independent biological replicate ($n = 3$). Data were presented as mean ± standard deviation. (E) RNA-seq heatmap showing expression of primed, naïve, general pluripotency, and 8-cell stage (8C) genes in the indicated cell populations under primed, naïve, or 4CL conditions. Each column represents an independent biological replicate ($n = 2$ per group). (F) Schematic of re-priming experiment showing MER51B and LTR5_Hs reporter expression (left panel). Representative flow cytometry plots showing changes in MER51B-GFP and LTR5_Hs-RFP expression during re-priming (naïve-to-primed conversion of UCLA4 cells) (right panel). (G) Schematic showing embryoid body formation assay (left panel). Flow cytometric quantification of MER51B-GFP and LTR5-Hs-RFP reporter expression during embryoid body formation (right panel). Each data point represents an independent biological replicate ($n = 3$). (H) Representative flow cytometry plots showing changes in MER51B-GFP and LTR5_Hs-RFP expression during UCLA4 embryoid body formation. (I) qRT-PCR analysis for *NANOG* and *OCT4*, 4 or 12 days after initiating the embryoid body formation assay shown in Fig. 2G, H. Each data point represents an independent biological replicate ($n = 3$). Statistical significance was determined using one-way analysis of variance (ANOVA) with Dunnett's post hoc correction and exact $p$ values are represented in the figure. Data were presented as mean ± standard deviation. Relative gene expression was normalized to *ACTB*. (J) Schematic of hESC monolayer differentiation assay using MEKi. (K) Flow cytometric quantification of MER51B-GFP and LTR5_Hs-RFP expression during UCLA4 monolayer differentiation using MEKi. Each data point represents an independent biological replicate ($n = 3$). (L) RNA-seq heatmap showing expression levels of selected genes associated with pluripotency, ectoderm, endoderm, and mesoderm in primed and differentiated UCLA4 cells after 6 days of MEKi treatment. Each column represents an independent biological replicate ($n = 2$ per group). (M) Schematic of overexpression of the transcription factor NGN2 on hESC to induce differentiation into neural progenitors (NPCs) (left panel). Representative flow cytometry plots showing changes in MER51B-GFP and LTR5_Hs-RFP expression following overexpression of NGN2 (right panel). Source data are available online for this figure.

## Concurrent expression of LTR5_Hs and MER51B marks a metastable population of cells and may reflect a DNA damage response

While characterizing dual reporter hESCs cultured under primed conditions, we unexpectedly identified several colonies containing 1–5 cells that were strongly labeled with both RFP and GFP, among cells expressing MER51B-GFP exclusively (Fig. EV3A). Flow cytometry further confirmed the presence of a rare GFP$^+$/RFP$^+$ (~0.3%) subpopulation within the primed cell population (Fig. 3A,B). We then investigated whether these unexpected primed cells, which co-expressed MER51B-GFP and LTR5_Hs-RFP, represented a stable cell population or if hPSCs transitioned into and out of the double-positive state. To this end, we sorted GFP$^+$ and GFP$^+$/RFP$^+$ cells for further culture under conventional primed conditions (Figs. 3C,D and EV3B). Upon culturing these purified subpopulations, we observed that GFP$^+$/RFP$^+$ cells produced GFP$^+$/RFP$^-$ cells and vice versa (Figs. 3D and EV3B). Remarkably, within four days, nearly all double-positive cells had reverted to a predominantly GFP$^+$/RFP$^-$ cell population, with only rare GFP$^+$/RFP$^+$ cells. Thus, the concurrent expression of LTR5_Hs-RFP and MER51B-GFP marks a rare, metastable population of primed hPSCs.

To further characterize cells co-expressing MER51B-GFP and LTR5_Hs-RFP within primed cultures, we sorted GFP$^+$/RFP$^+$ cells and GFP$^+$/RFP$^-$ cells and assessed changes in chromatin accessibility using ATAC-seq. GFP$^+$/RFP$^+$ cells showed increased chromatin accessibility at LTR5_Hs regions compared to GFP$^+$/RFP$^-$ primed cells (Fig. 3E). These data indicate that our reporter system effectively captures the chromatin accessibility state of MER51B and LTR5_Hs genomic elements. By contrast, we did not observe significant global changes in chromatin accessibility, suggesting that the two cell populations are not markedly different at the overall chromatin level (Fig. EV3C).

We further characterized the transcriptome of sorted GFP$^+$/RFP$^+$ and GFP$^+$/RFP$^-$ cells using RNA-seq (Figs. 3F–I and EV3D). GFP$^+$/RFP$^+$ cells expressed higher levels of LTR5_Hs transcripts than GFP$^+$/RFP$^-$ cells (Fig. 3F). Unbiased PCA analysis revealed

that GFP$^+$/RFP$^+$ and GFP$^+$/RFP$^-$ cells clustered together and segregated from GFP$^-$/RFP$^+$ naïve cells and dual-positive 4CL-converted cells (Fig. 3G). These data suggest that primed cells co-expressing MER51B-GFP and LTR5_Hs-RFP are transcriptionally similar to their primed GFP$^+$/RFP$^-$ counterparts. Indeed, cells expressing both reporters had only 327 transcripts activated more than twofold, and only 37 genes repressed more than twofold compared to GFP-only-expressing cells. Pathway analysis revealed that upregulated transcripts were associated with preimplantation embryo development and the p53 pathway (Fig. 3H). Among the genes highly enriched in GFP$^+$/RFP$^+$ cells were ZGA-related factors (De Iaco et al, 2017), including *H3Y1, MBD3L2/3, ZSCAN4*, and *KLF17* (Fig. 3I). Of note, the expression of ZGA-related transcripts was significantly lower than in TPRX1$^+$ 8-cell like cells (8CLCs) and 4CL-cultured GFP$^+$/RFP$^+$ cells (Figs. 3J and EV2D), indicating that these transcriptional changes do not represent a complete cell fate transition. Supporting this observation, flow cytometry data showed that dual-positive primed hPSCs exhibited lower LTR5_Hs and higher MER51B expression compared to cells in 4CL conditions (Fig. EV3E), suggesting that ZGA-related gene expression correlates with reporter intensity. Further inspection of our transcriptomic data revealed expression of downstream targets of the tumor suppressive transcription factor p53, such as *CDKN1A* (p21) and *PLK2* (Fischer, 2017) (Fig. 3I,K). We confirmed increased transcript levels for these genes using qRT-PCR in sorted GFP$^+$/RFP$^+$ from two hESC lines (Figs. 3L and EV3F), and an increase at the protein level for H3Y (Resnick et al, 2019) via immunofluorescence (Fig. 3M). The presence of a rare population expressing H3Y within primed cells (~0.2%) was further confirmed by generating a novel hPSC knock-in line expressing mCherry fluorescence at the 3' end of the *H3Y1* locus (Fig. EV3G). Thus, our dual reporter system identifies a rare population of primed cells that, while transcriptionally similar to conventional primed cells, transiently activate the p53 pathway and express low levels of early embryo-related transcripts.

The transcriptional similarity between GFP$^+$ and GFP$^+$/RFP$^+$ primed cells suggests that activation of LTR5_Hs and low-level

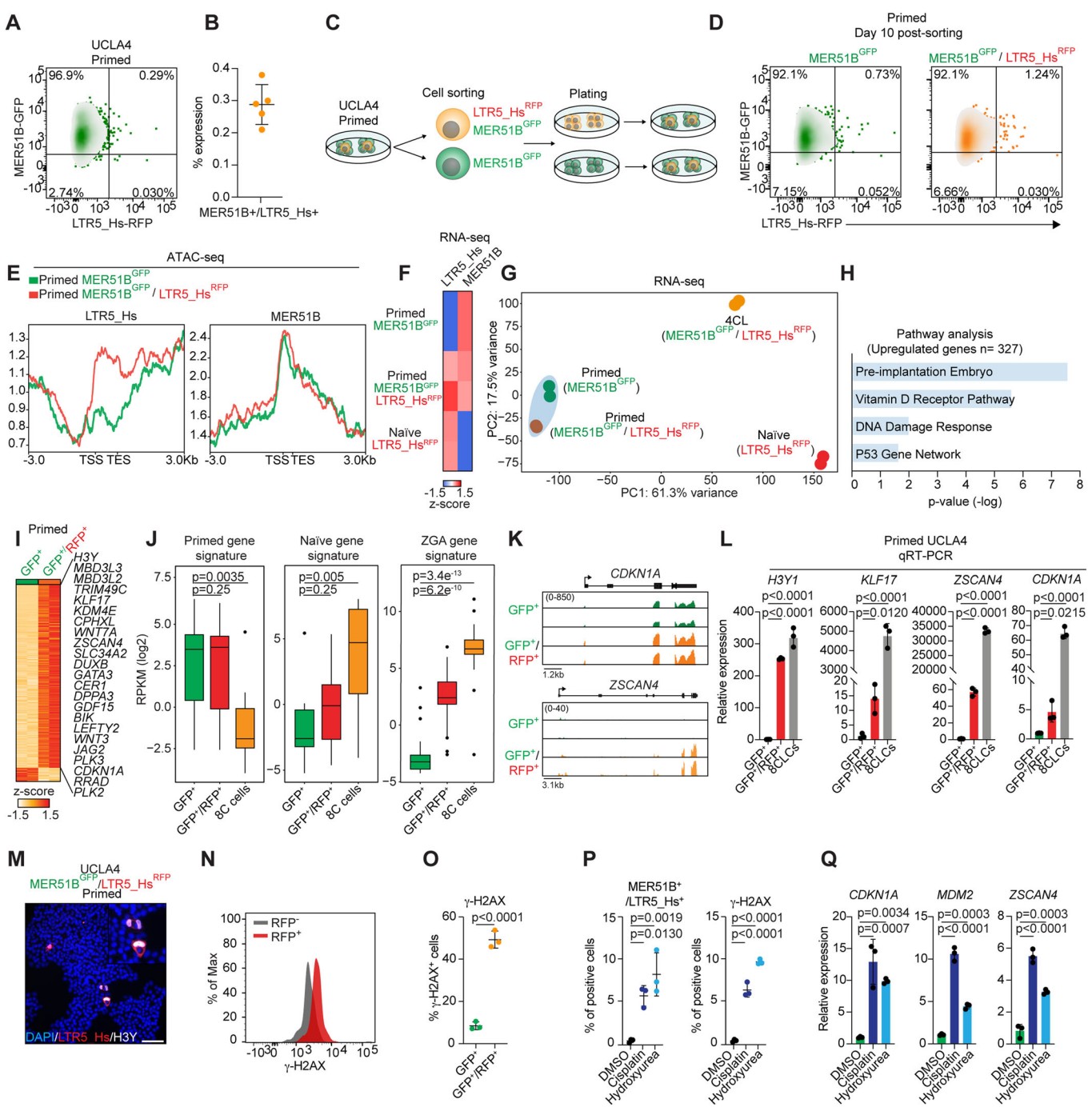

expression of ZGA-related genes are not reflective of cell fate change. We, therefore, asked why these genes would be activated in a small population of cells in the primed state. Recent reports have shown that DNA damage-induced activation of p53 can, in turn, activate DUX expression in a subset of human and mouse PSCs, leading to the transient expression of transcripts related to preimplantation development, including *ZSCAN4* and *H3Y1* (Grow et al, 2021). Similar results have also been reported in cancer cells, where a subset of metastable cancer cells transiently activates *H3Y1* and suppresses antigen presentation (Smith et al, 2023). Additionally, it is well known that p53 can directly bind to and activate

LTR5_Hs elements (Liu et al, 2022). A trivial explanation for our observations is that the fluctuating, double-positive cells might reflect a small population of cells responding to DNA damage. We first used flow cytometry to examine γ-H2AX levels, indicative of DNA double-strand breaks (Kuo and Yang, 2008), in primed GFP+ and GFP+/RFP+ cells. Notably, GFP+/RFP+ primed cells showed heightened γ-H2AX+ levels, with a tenfold increase in the number of γ-H2AX+ cells (~50%) compared to GFP+ only primed cells (~5%) (Fig. 3N,O). This finding supports our hypothesis that the metastable double-positive state reflects a DNA damage response. Importantly, GFP+/RFP+ cells in 4CL conditions, which reflect a

**Figure 3. MER51B and LTR5_Hs activity labels a rare, metastable subpopulation within primed hESCs.**

(A) A representative flow cytometry plot showing primed UCLA4 cells labeled by MER51B-GFP alone and those labeled by MER51B-GFP and LTR5_Hs-RFP. (B) Percentage of cells co-expressing MER51B-GFP and LTR5_Hs-RFP amongst primed UCLA4 cells. Each data point represents an independent biological replicate ($n = 5$). Data were presented as mean ± standard deviation. (C) Schematic showing sorting and replating of MER51B-GFP$^+$ and MER51B-GFP$^+$LTR5_Hs-RFP$^+$ cells. (D) Representative flow cytometry plots showing sorted MER51B-GFP$^+$ and MER51B-GFP$^+$LTR5_Hs-RFP$^+$ cells after 10 days in culture. (E) Chromatin accessibility read enrichment around MER51B and LTR5_Hs elements in sorted MER51B-GFP$^+$ and MER51B-GFP$^+$LTR5_Hs-RFP$^+$ primed UCLA4 cells. (F) Heatmap showing MER51B and LTR5_Hs expression in sorted MER51B-GFP$^+$, MER51B-GFP$^+$LTR5_Hs-RFP$^+$ primed and LTR5_Hs-RFP$^+$ naïve UCLA4 cells. Each row represents an independent biological replicate ($n = 2$ per group). (G) Unbiased PCA analysis of RNA-seq data for the indicated samples. Each data point represents an independent biological replicate ($n = 2$ per group). (H) Pathway analysis of upregulated genes (FC >2; $p$ value <0.05; FDR <0.05) in primed UCLA4 MER51B-GFP$^+$LTR5_Hs-RFP$^+$ versus MER51B-GFP$^+$ cells. Statistical significance was determined by two-tailed Fisher's exact test. (I) Heatmap showing expression of the indicated genes in primed MER51B-GFP$^+$LTR5_Hs-RFP$^+$ versus MER51B-GFP$^+$ UCLA4 cells. Each column represents a independent biological replicate ($n = 2$ per group). A gene was considered to be differentially expressed when abs(log$_2$(fold change)) >1, false discovery rate (FDR) <0.01 and $p$ value <0.05. Statistical significance was determined by Wald test with Benjamini–Hochberg correction. (J) Expression of primed ($n = 12$, Messmer et al, 2019), naïve ($n = 12$, Messmer et al, 2019), and ZGA ($n = 23$, Taubenschmid-Stowers et al, 2022) genes in primed UCLA4 MER51B-GFP$^+$ cells, MER51B-GFP$^+$LTR5_Hs-RFP$^+$ cells, and 8-cell stage cells (TPRX1$^+$) from (Data ref: Mazid et al, 2022). The box represents the interquartile range (IQR), spanning from the 25th percentile (lower bound) to the 75th percentile (upper bound). The horizontal line within the box indicates the median (50th percentile). Whiskers extend to the smallest and largest values within 1.5 × IQR from the lower (Q1) and upper (Q3) quartiles, respectively. Data points outside this range are plotted as outliers. Statistical significance was determined by unpaired two-tailed Student's $t$-test, and exact $p$ values are represented in the figure. $P$ values >0.05 are non-significant. (K) Gene tracks of RNA-seq data showing individual mRNAs enriched in primed MER51B-GFP$^+$LTR5_Hs-RFP$^+$ versus MER51B-GFP$^+$ UCLA4 cells. Each track represents an independent biological replicate. (L) qRT-PCR for the indicated genes in sorted primed MER51B-GFP$^+$LTR5_Hs-RFP$^+$ and 8CLCs (TPRX1$^+$) versus MER51B-GFP$^+$ UCLA4 cells. Each data point represents an independent biological replicate ($n = 3$). Statistical significance was determined by unpaired two-tailed Student's $t$-test, and exact $p$ values are represented in the figure. Data were presented as mean ± standard deviation. Relative gene expression was normalized to *ACTB*. (M) Representative immunofluorescence image of H3Y and LTR5_Hs-RFP in bulk UCLA4 reporter cells. Scale bar: 300 μm. (N) Representative flow cytometry plot of γ-H2AX staining intensity in LTR5_Hs-positive versus negative cells. (O) Flow cytometric quantification for the percentage of γ-H2AX$^+$ cells in gated MER51B-GFP$^+$LTR5_Hs-RFP$^+$ versus MER51B-GFP$^+$ primed UCLA4 cells. Each data point represents an independent biological replicate ($n = 3$). Statistical significance was determined by unpaired two-tailed Student's $t$-test, and exact $p$ values are represented in the figure. Data were presented as mean ± standard deviation. (P) Flow cytometry quantification of the percentage of MER51B-GFP$^+$LTR5_Hs-RFP$^+$ (left) and γ-H2AX$^+$ (right) cells after cisplatin and hydroxyurea treatment versus DMSO control. Each data point represents an independent biological replicate ($n = 3$). Statistical significance was determined using one-way analysis of variance (ANOVA) with Dunnett's post hoc correction, and exact $p$ values are represented in the figure. Data were presented as mean ± standard deviation. (Q) qRT-PCR for the indicated genes in bulk-primed cells treated with cisplatin or hydroxyurea for 10 and 12 h, respectively. Each data point represents an independent biological replicate ($n = 3$). Statistical significance was determined using one-way analysis of variance (ANOVA) with Dunnett's post hoc correction, and exact $p$ values are represented in the figure. Data were presented as mean ± standard deviation. Relative gene expression was normalized to *ACTB*. Source data are available online for this figure.

genuine cell fate change, showed no elevation in γ-H2AX levels (Fig. EV3H).

We then tested whether DNA damage could activate LTR5-Hs and ZGA transcripts in primed cells. To this end, we treated our dual reporter hPSC line with cisplatin and hydroxyurea, two chemical agents that induce double-stranded DNA breaks and activate the p53-mediated DNA damage response (Grow et al, 2021). Consistent with our interpretation, cisplatin, and hydroxyurea significantly expanded the fraction of primed cells co-expressing MER51B-GFP and LTR5_Hs-RFP, which correlated with the appearance of γ-H2AX-positive cells (Figs. 3P and EV3I,J). Moreover, treatment with cisplatin and hydroxyurea induced the expression of *ZSCAN4*, *CDKN1A*, and *MDM2*, as determined by qRT-PCR in bulk cultures (Fig. 3Q). Finally, we treated the cells with doxorubicin, a chemical that activates the p53-mediated DNA damage response as well as *ZSCAN4* expression (Fischer, 2017; Grow et al, 2021; Nakai-Futatsugi and Niwa, 2016; Storm et al, 2014). Doxorubicin exhibits intrinsic red fluorescence, emitting at 595 nm (Shah et al, 2017), thus precluding quantification of the LTR5_Hs-RFP reporter. Yet, our qRT-PCR analyses of unsorted treated cells indicate that doxorubicin treatment also leads to the activation of *ZSCAN4*, *CDKN1A*, and *MDM2* (Fig. EV3K), aligning with previous reports (Grow et al, 2021; Smith et al, 2023; Storm et al, 2014). Thus, DNA damage triggers a subpopulation of primed hPSCs to express transcripts associated with preimplantation development.

Together, our data reveals a rare, metastable primed cell population expressing low levels of transcripts associated with preimplantation human development and activity of p53.

## The LTR5_Hs and MER51B reporter systems facilitate the characterization of key regulators of cell potency

The tractability of our dual reporter system makes it ideal for uncovering key regulators of embryonic cell fate transitions. We first tested whether reporter dynamics could capture changes in hPSC fate induced by the manipulation of transcription factors. To this end, we converted primed hPSCs to a naïve state by overexpressing either KLF17 or the transcription factors NANOG and KLF2. These factors have been previously shown to induce naïve gene expression in primed cells (Lea et al, 2021; Takashima et al, 2014). Notably, KLF17 overexpression in primed culture conditions forced cells to exit the primed state, as highlighted by the loss of MER51B-GFP expression, with cells shifting toward an exclusive LTR5_Hs-RFP$^+$ state within 6 days (Fig. 4A). Similar results were obtained upon induction of NANOG and KLF2 (Fig. 4A).

We then hypothesized that our dual reporter system could facilitate the study of factors that have not been previously investigated in pluripotency maintenance and during the primed-to-naïve transitions. To test this hypothesis, we analyzed a previous loss-of-function screen conducted in hPSCs (Collier et al, 2022). This screen was designed to identify factors crucial for safeguarding primed pluripotency, and when knocked out, would lead to an accelerated primed-to-naïve transition. The vast majority of the screen hits remained unexplored. From the screen, we selected seven factors whose perturbation in primed cells is predicted to induce a naïve gene expression signature (Fig. 4B). Among these genes were known pluripotency regulators (e.g., FOXH1 (Wang et al, 2019) and ZIC3 (Yang et al, 2019)), as well as factors with

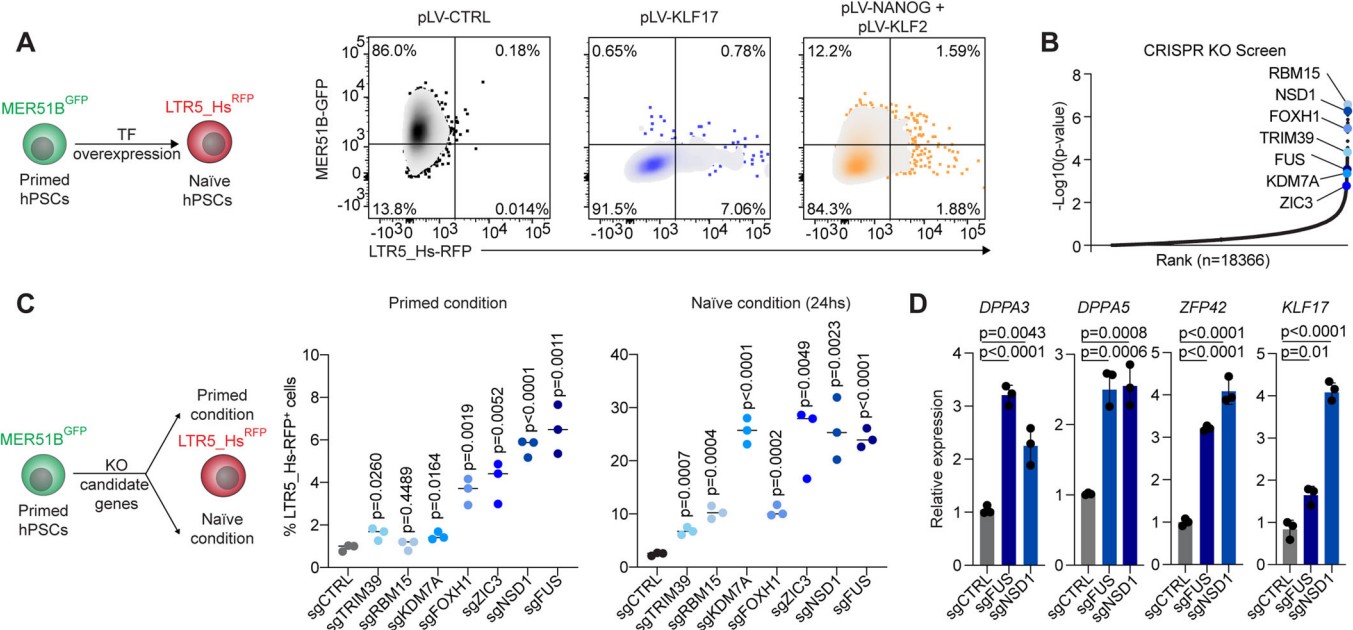

**Figure 4. MER51B and LTR5_Hs activity facilitates the study of regulators of stem cell potency.**

(A) Schematic representing the overexpression of transcription factors on primed hESC (MER51B-GFP$^+$) and the expected naïve (LTR5_Hs-RFP$^+$) phenotype outcome (left panel). Representative flow cytometry plots showing changes in MER51B-GFP and LTR5_Hs-RFP expression in primed UCLA4 cells upon overexpression of the transcription factors KLF17, or KLF2 and NANOG (right panel). (B) Ranking plot showing selected top hits from a genome-wide CRISPR screen for genes that act as barriers for the primed-to-naïve conversion (Collier et al, 2022). Statistical analysis for CRISPR screen data was performed using MAGECK (Model-based Analysis of Genome-wide CRISPR-Cas9 Knockout), which employs a negative binomial model to identify positively or negatively selected genes. Adjusted $p$ values (false discovery rate, FDR) were calculated to account for multiple testing. (C) Schematic representing the CRISPR knockout of the selected screening hits on primed hESC (MER51B-GFP$^+$) and the expected (LTR5_Hs-RFP$^+$) phenotype outcome (left panel). Percentage of LTR5_Hs-RFP$^+$ cells 6 days after CRISPR knockout of the indicated genes in primed culture conditions (left panel) or after 1 day of culture in naïve conditions (right panel). Each data point represents an independent biological replicate ($n = 3$), with the mean value shown as a horizontal line. Statistical significance was determined by unpaired two-tailed Student's $t$-test, and exact $p$ values are represented in the figure. $P$ values >0.05 are non-significant. (D) qRT-PCR for the indicated genes in the indicated bulk samples. Cells were cultured in 5i/LAF for 4 days following gene knockout. Each data point represents an independent biological replicate ($n = 3$). Statistical significance was determined by unpaired two-tailed Student's $t$-test, and exact $p$ values are represented in the figure. Data were presented as mean ± standard deviation. Relative gene expression was normalized to $ACTB$. Source data are available online for this figure.

unknown roles in pluripotency, including the chromatin regulators NSD1 (Sun et al, 2023) and KDM7A (Qu et al, 2023), the RNA-binding proteins FUS (Patel et al, 2015) and RBM15 (Cao et al, 2024), and the E3 ubiquitin-protein ligase TRIM39 (Zhang et al, 2012). To investigate the potential role of these factors in primed hPSCs, we performed CRISPR-mediated knockout in our dual reporter cells and tracked the emergence of LTR5_Hs-RFP$^+$ cells and the loss of MER51B-GFP signal over time. Remarkably, the depletion of FUS and NSD1 accelerated the reprogramming of primed hESCs to a naïve state in both primed and naïve culture conditions, characterized by increased induction of LTR5_Hs-RFP and elevated levels of the naïve markers $DPPA3$, $DPPA5$, $ZFP42$, and $KLF17$, (Figs. 4C,D and EV4A). In each case, expression of MER51B was silenced, indicating a bona fide cell fate transition, distinct from the double-positive cells induced by DNA damage. Furthermore, the depletion of FUS and NSD1 led to the generation of stable and homogenous GFP$^-$/RFP$^+$ naïve hPSC cultures within three passages (Fig. EV4B). These results identify factors with a key role in safeguarding primed pluripotency.

In conclusion, these data suggest that the LTR5_Hs-RFP and MER51B-GFP dual reporter system provides a valuable tool for studying critical regulators of cell potency and embryonic cell fate transitions.

During early embryo development, transposable elements are expressed in a stage-specific manner and contribute to the establishment of totipotency and pluripotency (DiRusso and Clark, 2023; Fueyo et al, 2022; Gifford et al, 2013; Goke et al, 2015; Hackett et al, 2017; Malik and Wang, 2022; Peaston et al, 2004; Rodriguez-Terrones and Torres-Padilla, 2018; Wang et al, 2014). For example, the transposon MuERV-L has been shown to be essential for preimplantation development (Sakashita et al, 2023), and its expression is widely used to track totipotent cells in vitro (Borsos and Torres-Padilla, 2016; Macfarlan et al, 2012). However, due to evolutionary differences in the repetitive elements present in the genome of mice and primates (Rodriguez-Terrones and Torres-Padilla, 2018), it remains unclear whether the stage-specific expression of primate-specific transposable elements can be used to define cell potency during human developmental transitions.

In this study, we defined the transposable elements MER51B and LTR5_Hs as sensitive, accurate indicators of human pluripotent stem cell states. Specifically, the transposon-based dual reporter system that we have introduced here robustly marks hPSCs in the primed and naïve states. Moreover, this system continuously tracks the dynamics of hPSC state transitions. Using this system, we were able to identify a rare, metastable cell population within primed cells expressing genes associated with preimplantation embryo

development and the DNA damage response. Additionally, we assessed determinants of the primed-to-naïve conversion, highlighting the potential use of our dual reporter system for future studies aimed at identifying developmental regulators.

For the first time, we identified MER51B as a transposable element specifically active in the human primed pluripotent cell state. MER51B belongs to the long terminal repeat family, specifically within the endogenous retrovirus group (Kojima, 2018). Our study revealed that MER51B activity might serve as an accurate predictor of the primed pluripotency state compared to conventional reporter systems that rely upon pluripotency genes that are expressed in both the primed and naïve states. Perturbations of the primed state, either by altering culture conditions or manipulating developmental regulators, led to significant changes in MER51B activity. Conversely, LTR5_Hs, a specific subtype of LTR sequences associated with HERV-K elements (Grow et al, 2015), is activated in the naïve state and silenced during the naïve-to-primed transition. Our work shows that coupling LTR5_Hs with MER51B can provide a robust readout of cell state changes during embryonic cell fate transitions. Future studies are needed to decipher how these transposable elements are regulated at the molecular level.

Using our system, we identified a rare, metastable cell population that fluctuates in primed cell culture and is expanded upon induction of DNA damage. Similar results were recently obtained in cancer cells (Smith et al, 2023). Though these rare cells express low levels of genes associated with ZGA and preimplantation embryo development, they are otherwise not dramatically different at the transcriptional and chromatin levels from conventional primed cells, indicating that they are not bona fide totipotent-like cells. Accordingly, the expression levels of ZGA-related factors in this metastable cell population are significantly lower than in 8-cell-like cells, which may explain why these changes were not detected in recent single-cell RNA sequencing analyses (Messmer et al, 2019; Yan et al, 2013). We hypothesize that the observed gene expression changes are triggered by activation of the transcription factor p53 and DNA damage response pathways. Indeed, our findings caution against the use of p53 activators or DNA damage-inducing agents in media to induce or culture 8C-like cells (Yu et al, 2022), as they might lead to superficial transcriptional activation of ZGA genes without concomitant acquisition of 8C-like cell potency. Further research will determine whether transposable elements serve as active DNA damage sensors or are inadvertently activated during the DNA damage response in stem cells and cancer.

The identification of a metastable state provides an interesting opportunity to study gene regulatory mechanisms that prevent inappropriate expression of early developmental genes in a later developmental context. Double-positive MER51B+/LTR5-Hs+ cells transiently express low levels of ZGA-associated transcripts but were not able to acquire a totipotent cell fate, suggesting a robust mechanism for rapidly suppressing their expression. Presumably, these regulatory mechanisms are important for preventing the inappropriate reversion of cells to the totipotent state. Our dual reporter system provides a valuable tool for future investigations into both the transcriptional and post-transcriptional barriers to totipotency, as well as the molecular mechanisms underlying this metastable state.

Finally, our reporter system facilitated the characterization of factors that play a role in safeguarding the human pluripotent cell state. Of particular interest are NSD1 and FUS. NSD1, a chromatin remodeler involved in H3K36 dimethylation, is important in post-implantation development in mice (Rayasam et al, 2003). Based on the role of H3K36 methylation during reprogramming to pluripotency (Hoetker et al, 2023; Serdyukova et al, 2023), we speculate that H3K36 methylation might reinforce primed pluripotency. Mutations in NSD1 could lead to destabilization of the transcriptional state of primed cells and subsequent reversion of the cells toward a naïve cell state. On the other hand, FUS is an RNA-binding protein that has emerged as a key repressor of heterochromatin in human cells (McCarthy et al, 2021). Heterochromatin regions are largely reorganized during the naïve-to-primed transition and early embryo development (Theunissen et al, 2016; Yang et al, 2019). It is, therefore, tempting to speculate that FUS-mediated establishment of heterochromatin in primed cells might be critical to safeguarding primed pluripotency.

In summary, our work identified transposable elements as key indicators of pluripotent cell states and demonstrated that our novel reporter system can identify pathways and factors that contribute to the regulation of pluripotency states. Future work is needed to identify molecular regulators of the identified transposable elements. Additionally, future studies will clarify the role of p53 in early development and the significance and regulation of preimplantation development transcripts in metastable stem and cancer cells.

## Methods

**Reagents and tools table**

| Reagent/resource | Reference or source | Identifier or catalog number |
| --- | --- | --- |
| **Experimental models** | | |
| UCLA4 hESC | Diaz Perez S.V et al, Hum. Mol. Genet, 2012 | RRID:CVCL_9955 |
| UCLA5 hESC | Diaz Perez S.V et al, Hum. Mol. Genet, 2012 | RRID:CVCL_9956 |
| UCLA4 LTR5_Hs-Turbo-RFP/MER51B | This study | |
| UCLA4 H3Y-mCherry | This study | |
| DR4 MEFs | Homemade | |
| **Recombinant DNA** | | |
| PB-LTR5_Hs-TK-TurboRED-Neomycin | Grow et al, 2015 | |
| PB-MER51B-TK-d2EGFP-Hygromycin | This study | |
| pLV-EF1α-IRES-Puro | Addgene | 85132 |
| pLV-EF1α-hKLF17-IRES-Puro | This study | |
| Psicor-EF1α -hKLF2-T2A- Puro | This study | |
| Psicor-EF1α-hNANOG-T2A-Puro | This study | |
| pSicoR-EF1a-T2A-Puro | This study | |

| Reagent/resource | Reference or source | Identifier or catalog number |
|---|---|---|
| LentiCRISPRv2-Puro with FE modification | Chen et al, 2013 | |
| LentiCRISPRv2-Puro with FE modification sgRBM15 | This study | |
| LentiCRISPRv2-Puro with FE modification sgNSD1 | This study | |
| LentiCRISPRv2-Puro with FE modification sgFOXH1 | This study | |
| LentiCRISPRv2-Puro with FE modification sgKDM7A | This study | |
| LentiCRISPRv2-Puro with FE modification sgZIC3 | This study | |
| LentiCRISPRv2-Puro with FE modification sgTRIM39 | This study | |
| LentiCRISPRv2-Puro with FE modification sgFUS | This study | |
| PBase mCherry Puro | This study | |
| pSpCas9 (BB)-2A-Neo | Addgene | 127762 |
| **Antibodies** | | |
| Mouse anti-human Thy-1 (CD90 APC/Fire-750) | BioLegend | RRID: AB_2734314 |
| CD75 (Purified anti-human CD75 (LN-6921) | BioLegend | RRID: AB_2194435 |
| Mouse anti-human γ-H2AX (Alexa Fluor 647) | BD Biosciences | RRID: AB_1645414 |
| Rat Pacific Blue™ anti-mouse CD90.2 (Thy-1.2) | BioLegend | RRID: AB_10641693 |
| Rat Anti-H3.X/Y antibody | Sigma-Aldrich | RRID: AB_2793533 |
| Rabbit Anti-RFP Antibody | Thermo Fisher | RRID: AB_2809945 |
| Rat anti-Mouse IgM Heavy Chain Alexa Fluor 647 conjugate | BioLegend | RRID: AB_2563478 |
| Goat anti-rat IgG (H + L) Alexa fluor 647 | Thermo Fisher | RRID: AB_141778 |
| Donkey anti-rabbit IgG (H + L) Alexa Fluor 594 | Thermo Fisher | RRID: AB_141637 |
| **Oligonucleotides and other sequence-based reagents** | | |
| sgRBM15 | GGTGAGGACTCGACTTCCCG | |
| sgNSD1 | TTGGATTGACCATTACCGAA | |
| sgFOXH1 | GATCATGGCCAAGTAGGTGT | |
| sgKDM7A | ATGAAGCGGTTCACGTCGTA | |
| sgZIC3 | CGAGATGCCCAACCGTGAGC | |
| sgTRIM39 | CGTCCAGTGGCACAACGGTG | |

| Reagent/resource | Reference or source | Identifier or catalog number |
|---|---|---|
| sgFUS | CCAGTCGAGCCATATCCCTG | |
| **Chemicals, enzymes, and other reagents** | | |
| Matrigel | Corning | 354277 |
| Geltrex | Thermo Fisher | A1413201 |
| TeSR-E8 | StemCell Technologies | 05990 |
| mTeSR1 | StemCell Technologies | 85850 |
| mTeSR1 Medium w/o Select Factors | StemCell Technologies | 05896 |
| DMEM | Corning | 10-013-CV |
| DMEM/F-12 | Sigma-Aldrich | D6421 |
| Advanced DMEM/F-12 | Gibco | 12634010 |
| Neurobasal medium | Thermo Fisher | 21103049 |
| AggreWell EB formation medium | StemCell Technologies | 05893 |
| KnockOut Serum Replacement (KSR) | Gibco | 10828028 |
| Opti-MEM | Gibco | 31985062 |
| MEM nonessential amino acid solution | Sigma-Aldrich | M7145 |
| L-Glutamine | Sigma-Aldrich | G7513 |
| Penicillin-Streptomycin | Sigma-Aldrich | P4333 |
| 2-Mercaptoethanol | Thermo Fisher | 31350010 |
| Bovine serum albumin solution (BSA) | Sigma-Aldrich | A8412 |
| Trypsin-EDTA | Sigma-Aldrich | T4049 |
| Accutase | Sigma-Aldrich | SF006 |
| DPBS | Sigma-Aldrich | D8537 |
| TrypLE | Thermo Fisher | 12604021 |
| Fetal bovine serum | Cytiva | SH30071.03 |
| N2 Supplement | Thermo Fisher | 17502048 |
| B27 Supplement | Thermo Fisher | 17504044 |
| bFGF | Peprotech | 100-18B |
| Activin A | Peprotech | 120-14E |
| rhLIF | Peprotech | 300-05 |
| Y-27632 | Axon Medchem | HY-10071 |
| IM-12 | Axon Medchem | 2511 |
| SB590885 | Axon Medchem | 2504 |
| WH-4-023 | Axon Medchem | 2448 |
| PD0325901 | Axon Medchem | 1408 |
| XAV939 | Axon Medchem | 1527 |
| Gö6983 | Axon Medchem | 2466 |
| TSA (Trichostatin A) | Selleck Chemicals | S1045 |
| DZNep | Selleck Chemicals | S7120 |
| IWR-1 | Sigma-Aldrich | I0161 |
| L-Ascorbic acid | Sigma-Aldrich | A4030 |
| G418 | InvivoGen | ant-gn-1 |
| Hygromycin | InvivoGen | ant-hg-1 |

| Reagent/resource | Reference or source | Identifier or catalog number |
|---|---|---|
| Puromycin | InvivoGen | ant-pr-1 |
| Doxorubicin | LC Laboratories | D400 |
| Cisplatin | Sigma-Aldrich | C2210000 |
| Hydroxyurea | Selleck Chemicals | S18960 |
| Restriction enzyme BsmBI-v2 | New England Biolabs (NEB) | R0580 |
| LunaScript RT Mix | New England Biolabs (NEB) | E3010 |
| Luna Universal qPCR Master Mix | New England Biolabs (NEB) | M3003 |
| SMART-Seq® v4 Ultra® Low Input RNA Kit for Sequencing | Takara Bio | 634894 |
| Nextera XT kit for Illumina | Illumina | |
| DAPI | Sigma-Aldrich | D9542 |
| IGEPAL® CA-630 | Alfa Aesar | J61055 |
| Illumina Tagment DNA TDE1 Enzyme and Buffer Kit | Illumina | 20034198 |
| **Software** | | |
| Prism10 | GraphPad | Version 10.2.2 |
| RStudio | RStudio Team (2020). RStudio: Integrated Development for R. RStudio, PBC, Boston, MA. | Version 2023.06.2 + 561 |
| **Other** | | |
| AggreWell 800 plate | StemCell Technologies | 34811 |
| Neon™ NxT Electroporation System 100-µL Kit | Invitrogen | N10025 |
| Monarch Total RNA Miniprep | New England Biolabs (NEB) | T2010S |
| miRNeasy Mini Kit | Qiagen | 217004 |
| High-Capacity RNA-to-cDNA Kit | Applied Biosystems | 4387406 |
| MiniElute Kit | Qiagen | 28004 |
| Fix and Perm Cell fixation and cell permeabilization Kit | Thermo Fisher | GAS003 |

## Human embryonic stem cells

Conventional human primed ESCs (UCLA4 and UCLA5) carrying MER51B-GFP and LTR5_Hs-RFP constructs and UCLA4 H3Y-mCherry were maintained on Matrigel (Corning)-coated dishes in E8 or mTeSR1 medium (StemCell Technologies) and passaged with Trypsin-EDTA solution (Sigma-Aldrich). For maintenance, cells were passaged every 3–4 days. Cells were confirmed negative for mycoplasma contamination weekly using a *Mycoplasma* PCR Detection Kit (Applied Biological Materials).

## 5i/LAF naïve cell culture

Naïve conversion of UCLA4 MER51B-GFP/LTR5_Hs-RFP hESC line was performed as previously described (Di Stefano et al, 2018). Briefly, primed cells were washed once with 1X DPBS (Sigma-Aldrich) and treated for 5 min with the TrypLE express enzyme (Thermo Fisher). Cells were dissociated into a single-cell suspension and plated at a density of 30,000 cells per 9.5 cm$^2$ on irradiated DR4 mouse embryonic fibroblasts (MEFs) in E8 medium supplemented with 10 µM Y-27632 (Axon Medchem) and 50 µg/mL of neomycin. Two days later, the medium was changed to 5i/LAF and then changed daily. 5i/LAF medium contained a 1:1 mixture of DMEM/F-12 (Sigma) and Neurobasal medium (Thermo Fisher) containing 1X N2 supplement (Thermo Fisher), 1X B27 supplement (Thermo Fisher), 10 ng/mL bFGF (Peprotech), 1% nonessential amino acids (Sigma-Aldrich), 1 mM L-glutamine (Sigma-Aldrich), penicillin-streptomycin (Sigma-Aldrich), 0.1 mM β-mercaptoethanol (Thermo Fisher), 50 µg/mL BSA (Sigma-Aldrich), 0.5 µM IM-12 (Axon Medchem), 0.5 µM SB590885 (Axon Medchem), 1 µM WH-4-023 (Axon Medchem), 10 µM Y-27632 (Axon Medchem), 20 ng/mL Activin A (Peprotech), 20 ng/mL rhLIF (Peprotech), 0.5% KSR (Gibco), and 1 µM PD0325901 (Axon Medchem). After roughly 8–10 days, cells were dissociated using Accutase (Sigma-Aldrich, SF006) and centrifuged in fibroblast medium [DMEM (Corning) supplemented with 10% FBS (Cytiva), 1 mM L-glutamine (Sigma-Aldrich), 1% nonessential amino acids (Sigma-Aldrich), penicillin-streptomycin (Sigma-Aldrich), 0.1 mM β-mercaptoethanol (Thermo Fisher)] and replated in 5i/LAF medium on irradiated DR4 MEFs after passing through a 40-µm cell strainer. Established naïve hESC lines were cultured on irradiated DR4 MEFs ($2.5 \times 10^6$ cells per 9.5 cm$^2$) in 5i/LAF medium and passaged every 6–7 days. Cells were fed daily with fresh medium. Naïve hESCs were cultured under low oxygen conditions (5% O$_2$). During naïve conversion, cells were detached as previously described on days 2, 4, and 6, as well as passages 3 (P3) and 4 (P4) for flow cytometry.

## PXGL naïve cell culture

Human naïve ESCs were also transitioned from 5i/LAF to PXGL (Bredenkamp et al, 2019) as previously described (Di Stefano et al, 2018). PXGL medium contained a 1:1 mixture of DMEM/F-12 (Sigma-Aldrich) and Neurobasal medium (Thermo Fisher) containing 1X N2 supplement (Thermo Fisher), 1X B27 supplement (Thermo Fisher), 10 ng/mL hLIF (Peprotech), 1% nonessential amino acids (Sigma-Aldrich), 1 mM L-glutamine (Sigma-Aldrich), penicillin-streptomycin (Sigma-Aldrich), 0.1 mM β-mercaptoethanol (Thermo Scientific), 50 µg/mL BSA (Sigma-Aldrich,), 1 µM or 0.5 µM PD0325901 (Axon Medchem), 2 µM XAV939 (Axon Medchem), 2 µM Gö6983 (Axon Medchem), 10 µM Y-27632 (Axon Medchem), 20 ng/mL human LIF (Peprotech). To establish the PXGL cell culture, human naïve ESCs previously cultured in 5i/LAF for at least three passages were passaged twice in PXGL. Naïve hESCs were cultured under low oxygen conditions (5% O$_2$).

## 4CL cell culture

Conversion and maintenance of naïve hESCs using a 4CL medium were performed as previously described (Mazid et al, 2022). 4CL medium consisted of a 1:1 mixture of Advanced DMEM/F-12

(Gibco) and Neurobasal medium (Thermo Fisher) supplemented with 1% N2 supplement (Thermo Fisher), 1% B27 supplement (Thermo Fisher), 1% nonessential amino acids, 1 mM L-glutamine, 1% penicillin-streptomycin, 1% sodium-pyruvate, 50 mg/mL L-ascorbic acid (Sigma-Aldrich), 20 ng/mL activin A (Peprotech), 20 ng/mL rhLIF (Peprotech), 1 μM PD0325901 (Axon Medchem), 5 nM TSA (Selleck Chemicals), 10 nM DZNep (Selleck), and 5 μM IWR-1 (Sigma-Aldrich). The medium was refreshed daily, and cells were passaged as single cells every 3–4 days; 5 μM Y-27632 was added to the medium for the first 24 h. Cells in 4CL were cultured at 37 °C, 5% $O_2$, and 5% $CO_2$. TPRX1$^+$ cells used for RNA extraction and qRT-PCR were obtained as described earlier (Mazid et al, 2022). During 4CL conversion, cells were detached for flow cytometry analysis on days 4, 12, and after P3.

## Re-priming

Re-priming of naïve hESCs was performed as previously described (Rostovskaya et al, 2022; Rostovskaya et al, 2019). Briefly, naïve cells were initially plated in feeder-free conditions on non-coated tissue culture-grade plates. Geltrex was added directly to the cells at a final 1 μL/cm² concentration. To dissociate the naïve cells, TrypLE was used, and the cells were then replated in Geltrex-coated wells at a seeding density of $1.6 \times 10^4$ cells/cm² in naïve medium supplemented with 10 μM Y-27632 (Axon Medchem). After 48 h, cells were washed with DMEM/F-12 supplemented with 0.1% BSA, and a re-priming medium (N2B27 with 2 μM XAV939) was added. The medium was replaced daily. Cells were passaged at a 1:2 ratio upon reaching confluency using TrypLE and 10 μM Y-27632. Following 10 days of culture, cells were transferred to Geltrex pre-coated plates in E8 medium. For passaging at this stage, cells were dissociated with 0.5 mM EDTA, and 10 μM Y-27632 was added for the first 24 h after each passage.

## Pluripotency exit assay

For the exit assay (Gonzales et al, 2015), 40,000 hESCs were seeded into each well of a Matrigel-coated 24-well plate in mTeSR1 medium (StemCell Technologies) supplemented with 10 μM Y-27632 (Axon Medchem). Twenty-four hours after seeding, mTeSR1 medium was replaced with the following differentiation media: -bFGF, -TGFβ condition (mTeSR1 medium without Select Factors (StemCell Technologies)) or MEK pathway inhibition (mTeSR1 supplemented with 2.5 μM PD0325901 (Axon Medchem)). Cells were incubated in differentiation media for 8 days for the -bFGF -TGFβ condition and 6 days for the MEKi condition. Cells were collected every 2 days for flow cytometry analysis and qRT-PCR.

## Embryoid body formation

Embryoid body (EB) formation was performed using an AggreWell 800 plate (StemCell Technologies) in AggreWell EB formation medium (StemCell Technologies). About $3 \times 10^6$ cells were seeded per AggreWell well to obtain ~300 EBs of ~10,000 cells each. Cells were maintained on AggreWell plates in EB Formation medium supplemented with 10 μM Y-27632 (Axon Medchem) for 24 h. EBs were harvested from AggreWell plates and maintained in ultralow adhesion plates in 5 mL AggreWell EB formation medium for

12 days. Media changes were performed every other day. EBs were harvested for RNA and flow cytometry analysis on days 4 and 12.

## DNA-damaging agent treatment

About 50,000 UCLA4 MER51B-GFP/LTR5_Hs-RFP cells were seeded into each well of a Matrigel-coated 24-well plate in E8 medium supplemented with 10 μM Y-27632 (Axon Medchem). Twenty-four hours after seeding, the medium was changed to E8 without Y-27632. After 48 h, the medium was switched to E8 supplemented with 1 μM of doxorubicin (LC Laboratories) for 8 h, 2 μM of cisplatin (Sigma-Aldrich) for 10 h, or 5 mM hydroxyurea (Selleck) for 12 h. Cells were analyzed by flow cytometry 12 h after washing off doxorubicin and hydroxyurea, and 40 h after cisplatin treatment.

## Vectors and cloning

The PB-LTR5_Hs-TK-TurboRED-Neomycin construct (Grow et al, 2015) was a kind gift from Dr. J. Wysocka. The PB-MER51B-TK-d2EGFP-Hygromycin construct was synthesized by VectorBuilder. The MER51B sequence corresponds to the following genomic region: chr11:94650010-94650508 (GRCh38). The overexpression construct for *KLF17* was generated using the pLV-Ef1α-IRES-Puro vector (Addgene, #85132), and pSicoR-Ef1a-T2A-Puro was used to overexpress *NANOG* and *KLF2*. pSicoR-EF1a-T2A-Puro was cloned in-house using the vector pSicoR-EF1a-mCh-Puro as a template (Addgene, #31845). The coding sequence (CDS) for the overexpressed genes was obtained by PCR amplification using targeted primers on UCLA4 hESC total cDNA. For CRISPR-based knockout experiments, sgRNA oligos were designed using IDT Cas9 gRNA design checker tool. sgRNAs were cloned into BsmBI-digested lentiCRISPRv2-puro with FE modification (Chen et al, 2013). SgRNA sequences are specified in the Reagents and Tools table.

## Cell line generation

About $2.5 \times 10^6$ UCLA4 and UCLA5 hESCs were co-transfected with 3.5 μg of PB-MER51B-TK-d2EGFP-Hygromycin, 3.5 μg of PB-LTR5_Hs-TK-TurboRED-Neomycin, and 1.4 μg of PBase (Vector-Builder) using the Neon Transfection System (Life Technologies). Electroporation parameters used were 1350 V, 30 ms, and 2 pulses. Stably transfected cells were selected with Neomycin and hygromycin 4 days after nucleofection during at least three passages. For knock-in H3Y-mCherry generation, the sgRNA targeting the insertion site of the donor construct (containing mCherry–puromycin cassette flanked by homologous arms) was cloned into the pSpCas9 (BB)-2A-Neo vector (Addgene). About $1 \times 10^6$ primed UCLA4 hESCs were electroporated using 2 μg of the donor construct and 2 μg of the sgRNA-Cas9 plasmid. Cells were seeded in E8 supplemented with 10 μM Y-27632 (Axon Medchem) for 4 days. Selection with 0.3 μg/mL puromycin was started after 48 h of nucleofection and maintained until cell death ceased.

## Virus production and transduction of hESCs

HEK293T cells were co-transfected with transfer plasmid and packaging plasmids (VSV-G and D8.9 for lentiviral production)

using calcium phosphate transfection. Viral supernatants were collected 24 h later and concentrated by ultracentrifugation at $21,000 \times g$ for 2 h at 4 °C. Concentrated viruses were resuspended in Opti-MEM (Gibco) and stored at −80 °C if not used immediately. Cells were transduced by adding the virus to the medium the day after passaging.

## Reversibility assay

UCLA4 MER51B-GFP/LTR5_Hs-RFP were sorted for GFP⁺ and GFP⁺/RFP⁺ expression on a FACSAria (BD Biosciences). Each cell population was pelleted by centrifugation at $500 \times g$ for 5 min and resuspended in E8 medium supplemented with 10 µM Y-27632. Cells were plated back in culture right after sorting on Matrigel-coated wells and analyzed by flow cytometry 4, 7, and 10 days after sorting.

## Knockout of screening hits and overexpression of pluripotency-associated transcription factors

About 150,000 primed UCLA4 MER51B-GFP/LTR5_Hs-RFP hESC were plated into each well of a Matrigel-coated 12-well plate in E8 medium supplemented with 10 µM Y-27632 (Axon Medchem). The day after, the medium was changed to E8 without Y-27632, and the cells were infected with each sgRNA vector or overexpressing vector. The cells were selected with 1 µg/mL of puromycin for 48 h. After 3 days of infection, 40,000 cells infected with each DNA construct were seeded into separate wells of Matrigel-coated 24-well plates in E8 medium supplemented with 10 µM Y-27632 (Axon Medchem). The day after, the medium was changed to E8. The following day, the medium was changed to E8 or 5i/LAF, and the cells were analyzed by flow cytometry 24 h after.

## RNA preparation

RNA for qRT-PCR was isolated from cells using a Monarch Total RNA Miniprep Kit (NEB) and reverse transcribed to cDNA using LunaScript RT Mix (NEB), according to the manufacturer's instructions. RNA for sequencing was isolated using the miRNeasy mini kit (Qiagen). RNA was eluted from the columns using RNase-free water and quantified using a Nanodrop ND-1000. cDNA was produced with the High-Capacity RNA-to-cDNA Kit (Applied Biosystems). To generate material for sequencing, cells were sorted based on GFP⁺ and GFP⁺/RFP⁺ expression and cell viability (DAPI staining).

## qRT-PCR analyses

qRT-PCR reactions were set up in triplicate with the Luna® Universal qPCR Master Mix (NEB). Real-time PCR was performed in 96-well PCR plates (Bio-Rad) on a CFX96 Real-Time PCR Detection System (Bio-Rad). Cycling conditions were as follows: 95 °C for 1 min, then 40 cycles of 95 °C for 15 s and 60 °C for 30 s. Primers are available upon request.

## RNA-seq

Total RNA was isolated from cells using the miRNeasy Micro Kit (Qiagen), according to the manufacturer's instructions.

Polyadenylated RNAs were enriched, and cDNA libraries were constructed using the SMART-Seq V4 ultra + Nextera XT kit for Illumina (Takara Bio). Libraries were sequenced on a HiSeq 4000 (Illumina) with paired-end 150 bp reads. For RNA-Seq of primed and naïve samples, MER51B-GFP⁺, and LTR5_Hs-RFP⁺ cells, respectively, were sorted before downstream analysis (Fig. 1F).

## ATAC-seq

ATAC-seq was performed as previously described (Buenrostro et al, 2013). About 50,000 cells were washed once with 100 µL PBS and resuspended in 50 µL lysis buffer (10 mM Tris-HCl pH 7.4, 10 mM NaCl, 3 mM MgCl₂, 0.2% IGEPAL). The suspension of nuclei was then centrifuged for 10 min at $500 \times g$ at 4 °C, followed by the addition of 50 µL transposition reaction mix (25 µL TD buffer, 2.5 µL Tn5 transposase, and 22.5 µL nuclease-free water) and incubation at 37 °C for 30 min. DNA was isolated using a MiniElute Kit (Qiagen), and libraries were amplified by PCR (13 cycles). After PCR, libraries were size-selected for fragments between 100 and 1000 bp with AmpureXP beads (Beckman Coulter). Libraries were purified with a QIAquick PCR Purification Kit (Qiagen), and integrity was checked on a Bioanalyzer before sequencing.

## Flow cytometry

The following antibodies were used: APC/Fire-750 mouse anti-human Thy-1/CD90 (1:200 dilution) (BioLegend cat. no. 328137), mouse purified anti-human CD75 (1:100 dilution) (BioLegend), Alexa Fluor 647 mouse anti-γ-H2AX (1:100 dilution) (BD Biosciences), rat anti-mouse IgM heavy chain Alexa Fluor 647 conjugate (1:1000 dilution) (BioLegend) and Pacific Blue rat anti-mouse CD90.2/Thy-1.2 (1:500 dilution) (BioLegend, used to exclude MEFs) were analyzed with an LSR-II (BD Biosciences), FACSCanto (BD Biosciences) or LSRFortessa (BD Biosciences) using FACSDiva v6.1.2 (BD Biosciences) and FlowJo software v10 (BD Biosciences). DAPI or DRAQ7 were used as a viability dye. For intracellular flow cytometry, cells were fixed and permeabilized using the FIX & PERM Cell Fixation and Permeabilization Kit (Thermo Fisher) according to the manufacturer's instructions. Rat anti-mouse IgM heavy chain Alexa Fluor 647 (1:1000 dilution) (Thermo Fisher) was used as the secondary antibody.

## Immunofluorescence

For immunostaining, the cells were fixed with 4% paraformaldehyde followed by 100% ice-cold methanol, blocked, and incubated with primary antibodies overnight at 4 °C. They were then stained with secondary antibodies donkey anti-rabbit IgG (H + L) (1:1000 dilution) (Alexa Fluor 594, Thermo Fisher) and goat anti-rat IgG (H + L) (1:1000 dilution) (Alexa Fluor 647, Thermo Fisher) at room temperature for 1 h. Nuclear staining was performed with DAPI (BD Biosciences). The following primary antibodies were used in this study: Rat anti-human H3.X/Y antibody (1:250 dilution) (Sigma-Aldrich), rabbit anti-RFP antibody (1:2500 dilution) (Thermo Fisher).

## Quantification and statistical analysis

Quantitative data are presented as mean ± s.d. unless otherwise indicated. Statistical analyses were performed using Prism software

(GraphPad). Details for statistical analyses, including replicate numbers, are included in figure legends.

## RNA-seq analysis

Raw sequencing files (FASTQ format) were aligned to the human reference genome (GRCh38) using STAR (version 2.5.1b) (Dobin et al, 2013) with default settings. Duplicates were removed using samtools (version 1.3.1) (Tarasov et al, 2015), and read counts for individual genes were generated using featureCounts (version 2.0.2) (Liao et al, 2014). Differential expression analysis was performed using R package DESeq2 (Love et al, 2014). Differentially expressed genes were defined by 2 FC, $p < 0.05$. Analysis of enriched functional categories among detected genes was formed using EnrichR (Xie et al, 2021). The 8C-like cell gene expression data (Mazid et al, 2022) were normalized using EdgeR (Robinson et al, 2010) and plotted using ggplot2.

## ATAC-seq analysis

Raw sequencing reads were mapped to the human reference genome or transgene sequence (GRCh38) using Bowtie2 (version 2.2.8). Reads not mapped to chromosomes, including those mapped to the inserted transgene sequences, were removed using samtools (version 1.3.1) (Tarasov et al, 2015), followed by peak calling with software MACS2 (version 2.2.9.1) (Zhang et al, 2008). Differentially accessible peaks were performed using R package csaw (version 1.28.0) (Lun and Smyth, 2016) and were defined as peaks with $p < 0.05$ and log2 fold change >1.5. The normalized bigwig files for visualizing read coverage over transposable elements were generated using Deeptools (version 3.5.4) (Ramirez et al, 2016).

# Data availability

The datasets produced in this study are available in the following database: RNA-seq and ATAC-seq data: Gene Expression Omnibus GSE272196 (https://www.ncbi.nlm.nih.gov/geo/query/acc.cgi?acc=GSE272196).

The source data of this paper are collected in the following database record: biostudies:S-SCDT-10_1038-S44319-024-00343-y.

# Peer review information

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

## Acknowledgements

We thank all members of the Di Stefano and Brumbaugh labs for stimulating discussions. We thank J. Wysocka for sharing the PB-LTR5_Hs vector. BDS is a Cancer Prevention and Research Institute of Texas (CPRIT) Scholar in Cancer Research. BDS is supported by the CPRIT Recruitment of First-Time, Tenure-Track Faculty Member Award RR200079, the American Society of Hematology (ASH) Scholar Award, the Andrew McDonough B+ Foundation (AMBF), the Worldwide Cancer Research (WCR) Foundation, and the NIH NIGMS MIRA award 1R35GM147126-01. SK is supported by NIH NCI 1F32CA288043-01. This project was supported by the Cytometry and Cell Sorting Core at Baylor College of Medicine with funding from the Cancer Prevention and Research Institute of Texas RP180672 and the NIH P30 CA125123 and S10 RR024574.

## Author contributions

**Florencia Levin-Ferreyra**: Data curation; Software; Formal analysis; Investigation; Methodology. **Srikanth Kodali**: Data curation; Formal analysis; Methodology. **Yingzhi Cui**: Data curation; Software; Visualization; Computational analyses. **Alison R S Pashos**: Formal analysis; Investigation; Methodology. **Patrizia Pessina**: Formal analysis; Investigation; Methodology. **Justin Brumbaugh**: Conceptualization; Supervision. **Bruno Di Stefano**: Conceptualization; Supervision; Funding acquisition; Visualization; Writing—original draft.

Source data underlying figure panels in this paper may have individual authorship assigned. Where available, figure panel/source data authorship is listed in the following database record: biostudies:S-SCDT-10_1038-S44319-024-00343-y.

## Disclosure and competing interests statement

The authors declare no competing interests.

# Expanded View Figures

**Figure EV1.  Differential transcriptional and epigenetic regulation of MER51B and LTR5_Hs in primed and naïve hESCs.**

(**A**) Heatmap of RNA-seq data (Data ref: Di Stefano et al, 2018) showing expression of TEs across different hESC lines under primed or conditions. Each column represents an independent biological replicate for primed ($n = 5$) or naïve conditions ($n = 14$). (**B**) Genomic heatmaps (ChIP-seq) showing H3K27ac, NANOG, and OCT4 levels at LTR5_Hs genomic regions in naïve and primed hESCs (Data ref: Chovanec et al, 2021). (**C**) Genomic heatmaps (ChIP-seq) showing SOX2 binding at LTR5_Hs genomic regions in naïve and primed hESCs (Data ref: Chovanec et al, 2021). (**D**) Gene tracks of specific genomic loci for LTR5_Hs and MER51B based on ChIP-seq data. (**E**) MER51B and LTR5_Hs expression during human preimplantation embryo development (Data ref: Xue et al, 2013). Violin plots show the distribution of read counts for each transposable element across stages of human preimplantation embryo development (zygote: $n = 2$, 2-cell: $n = 3$, 4-cell: $n = 3$, 8-cell: $n = 11$, morula: $n = 3$). The red dotted line indicates the median value, while the black dotted lines represent the interquartile range (IQR), spanning the 25th to 75th percentiles. (**F**) MER51B and LTR5_Hs chromatin accessibility during human preimplantation embryo development (Data ref: Liu et al, 2019). (**G**) Representative phase contrast and fluorescence microscopy images of MER51B-GFP/LTR5_Hs-RFP reporter UCLA4 cells in two different naïve state conditions (5i/LAF and PXGL). Scale bar 100 μm. (**H**) Correlation heatmap showing the Pearson correlation ($r$) values between UCLA4 MER51B-GFP/LTR5_Hs-RFP primed and naïve hESC RNA-seq samples. The scale bar represents the range of the correlation coefficients ($r$) displayed.

                                              

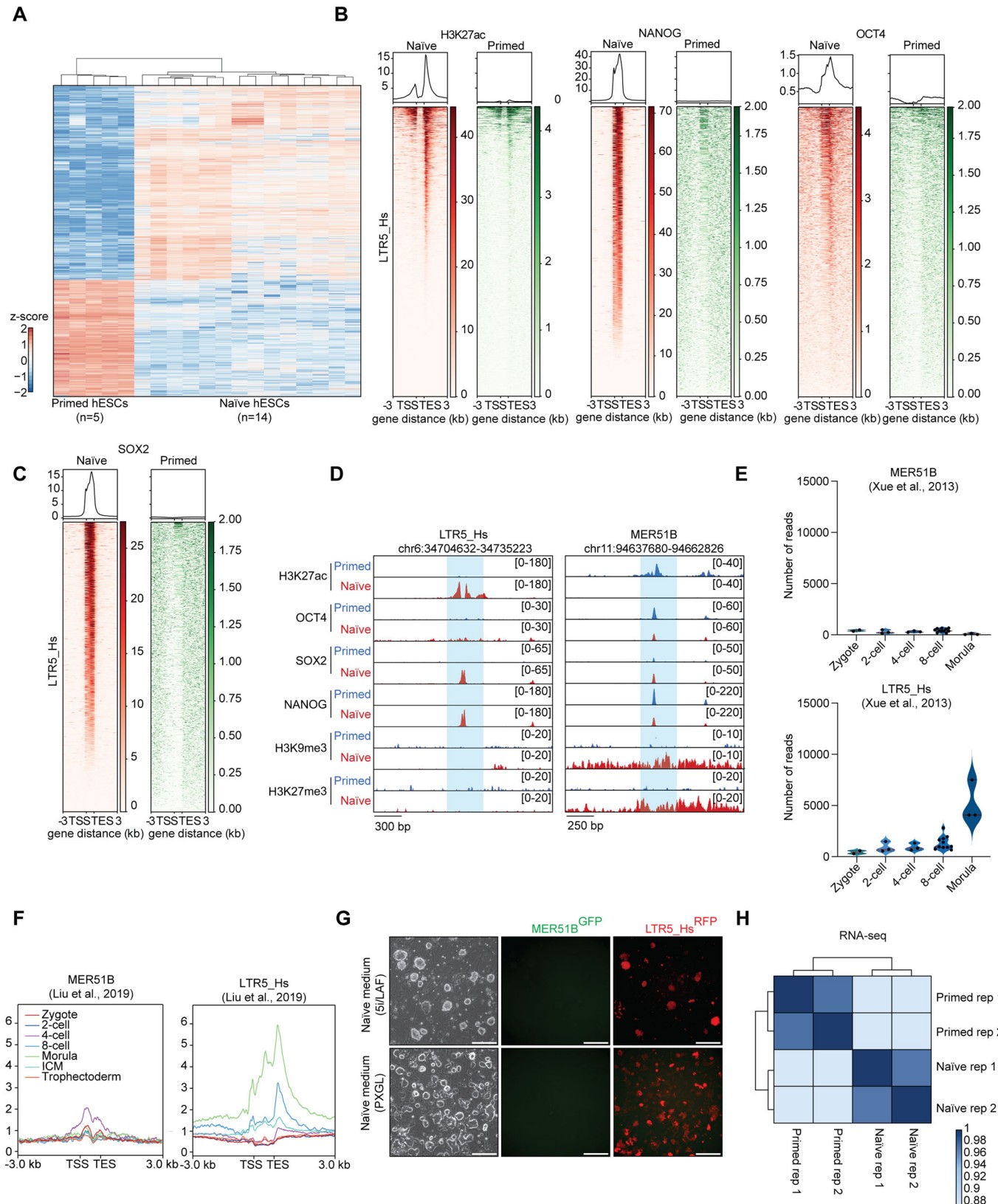

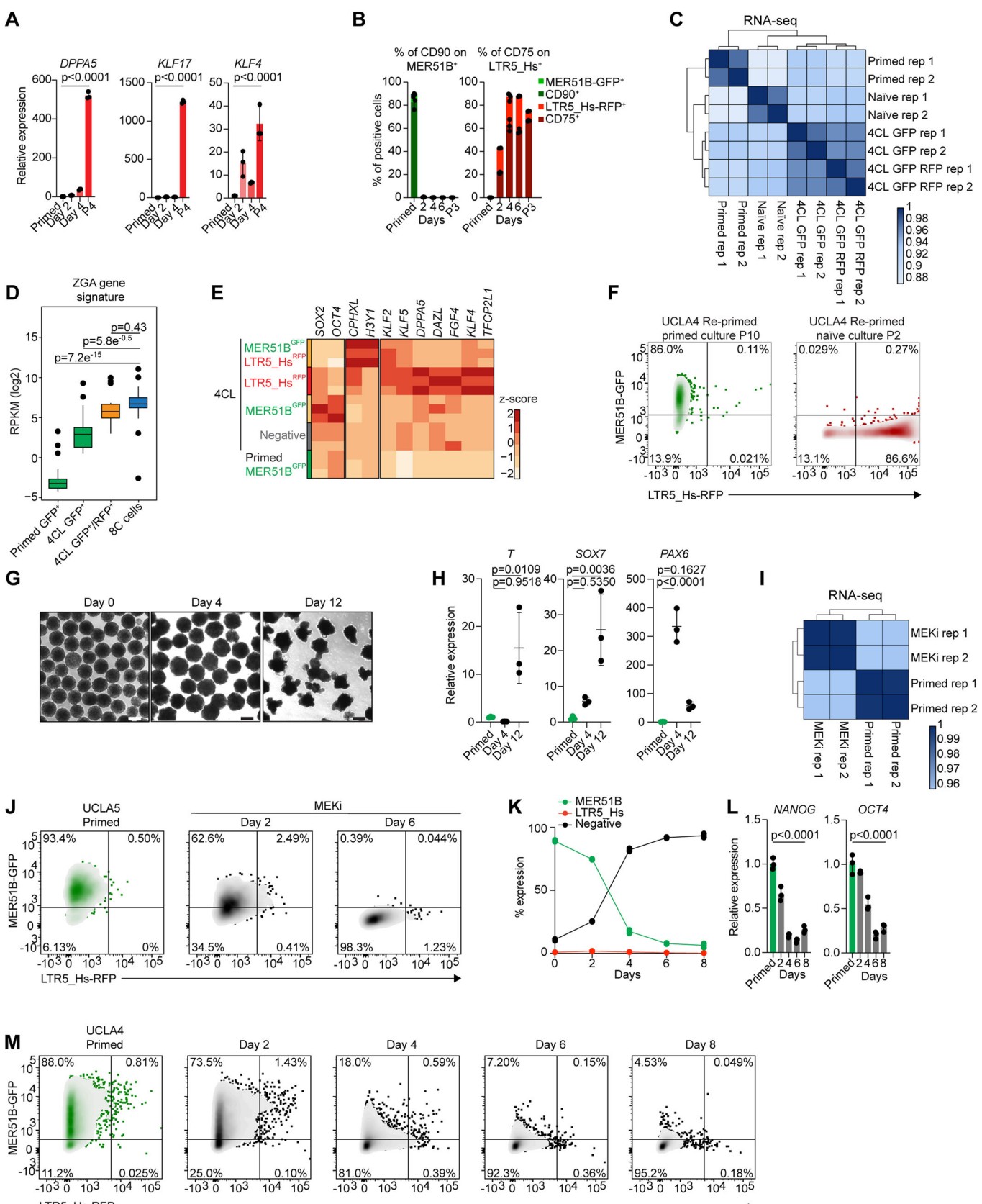

◀

**Figure EV2.  MER51B and LTR5_Hs activity can be used to monitor hESC conversion and differentiation dynamics.**

(A) qRT-PCR for the indicated genes during primed-to-naïve conversion at days 2 and 4 and at P4 of an established cell line. Each data point represents an independent biological replicate ($n = 3$). Statistical significance was determined using one-way analysis of variance (ANOVA) with Dunnett's post hoc correction, and exact $p$ values are represented in the figure. Data were presented as mean ± standard deviation. Relative gene expression was normalized to *ACTB*. (B) Flow cytometric analysis showing the percentage of MER51B-GFP$^+$ cells and CD75 expression in LTR5_Hs-RFP$^+$ cells during the primed-to-naïve transition. Each data point represents an independent biological replicate ($n = 3$). Data were presented as mean ± standard deviation. (C) Correlation heatmap showing the Pearson correlation ($r$) values between primed, naïve, and 4CL UCLA4 RNA-seq samples. The scale bar represents the range of the correlation coefficients ($r$) displayed. (D) Expression of ZGA ($n = 23$, Taubenschmid-Stowers et al, 2022) genes in primed UCLA4 MER51B-GFP$^+$ cells, 4CL UCLA4 MER51B-GFP$^+$ cells, MER51B-GFP$^+$LTR5_Hs-RFP$^+$ cells, and 8-cell (8C) stage cells (TPRX1$^+$) from (Data ref: Mazid et al, 2022). The box represents the interquartile range (IQR), spanning from the 25th percentile (lower bound) to the 75th percentile (upper bound). The horizontal line within the box indicates the median (50th percentile). Whiskers extend to the smallest and largest values within 1.5 × IQR from the lower (Q1) and upper (Q3) quartiles, respectively. Data points outside this range are plotted as outliers. Statistical significance was determined by unpaired two-tailed Student's *t*-test and exact $p$ values are represented in the figure. $P$ values >0.05 are non-significant. (E) Heatmap showing relative expression values of qRT-PCR for the indicated genes on sorted MER51B-GFP$^+$, MER51B-GFP$^+$LTR5_Hs-RFP$^+$, LTR5_Hs-RFP$^+$, and negative (MER51B-GFP$^-$LTR5_Hs-RFP$^-$) 4CL UCLA4 cells. Each row represents an independent biological replicate ($n = 3$). Relative gene expression was normalized to *HPRT*. (F) Representative flow cytometry plots showing changes in MER51B-GFP and LTR5_Hs-RFP expression during primed-to-naïve conversion of reprimed cells shown in Fig. 2F. (G) Representative phase contrast images of embryoid bodies at the indicated time points. Scale bar: 100 μm. (H) qRT-PCR analysis for *T*, *SOX7*, and *PAX6*, 4 or 12 days after initiating the embryoid body formation assay shown in Fig. 2G, H. Each data point represents an independent biological replicate ($n = 3$). Statistical significance was determined using one-way analysis of variance (ANOVA) with Dunnett's post hoc correction, and exact $p$ values are represented in the figure. $P$ values >0.05 are non-significant. Data were presented as mean ± standard deviation. Relative gene expression was normalized to *ACTB*. (I) Correlation heatmap showing the Pearson correlation ($r$) values between primed and differentiated (MEKi) UCLA4 RNA-seq samples after 6 days. The scale bar represents the range of the correlation coefficients ($r$) displayed. (J) Representative flow cytometry plots showing MER51B-GFP and LTR5_Hs-RFP expression during MEKi-induced UCLA5 cell differentiation. (K) Flow cytometric quantification of MER51B-GFP and LTR5_Hs-RFP expression during UCLA4 differentiation using primed hESC medium lacking TGF-β and bFGF. Each data point represents an independent biological replicate ($n = 3$). (L) qRT-PCR for the indicated genes during the hESC differentiation represented in Fig. EV2K. Each data point represents an independent biological replicate ($n = 3$). Statistical significance was determined using one-way analysis of variance (ANOVA) with Dunnett's post hoc correction, and exact $p$ values are represented in the figure. Data were presented as mean ± standard deviation. Relative gene expression was normalized to *ACTB*. (M) Representative flow cytometry plots showing MER51B-GFP and LTR5_Hs-RFP expression changes during UCLA4 differentiation using primed hESC medium lacking TGF-β and bFGF.

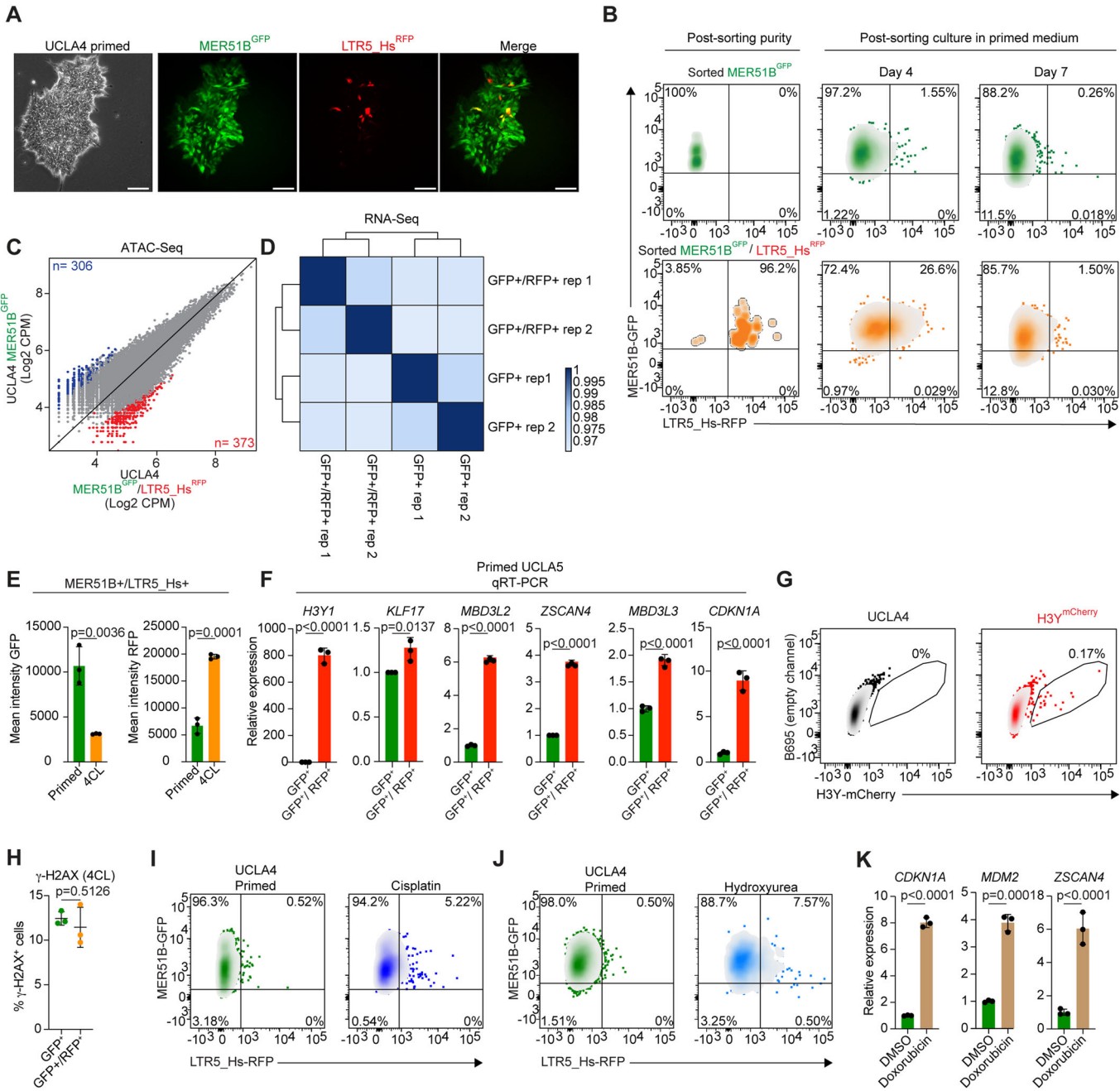

**Figure EV3.  MER51B-GFP and LTR5_Hs-RFP label a primed, metastable hESC population that expresses ZGA and DNA damage response genes.**

(A) Representative phase contrast and fluorescence microscopy images of UCLA4 MER51B-GFP/LTR5_Hs-RFP hESCs showing rare GFP$^+$RFP$^+$ (yellow) cells amongst the primed cell population. Scale bar: 100 µm. (B) Representative flow cytometry plots showing reporter expression changes in sorted MER51B-GFP$^+$ and MER51B-GFP$^+$LTR5_Hs-RFP$^+$ UCLA4 cells at the indicated time points after culture. (C) Scatterplot showing ATAC-seq analysis of chromatin accessibility for MER51B-GFP$^+$ and MER51B-GFP$^+$LTR5_Hs-RFP$^+$ UCLA4 cells. Two ($n = 2$) independent biological replicates per group were analyzed. Blue dots indicate genomic regions showing significantly decreased chromatin accessibility in MER51B-GFP$^+$LTR5_Hs-RFP$^+$ cells (Log2FC $< -1.5$, $p < 0.05$, $n = 306$); red dots indicate genomic regions showing significantly increased chromatin accessibility in MER51B-GFP$^+$LTR5_Hs-RFP$^+$ cells (Log2FC $> 1.5$, $p < 0.05$, $n = 373$). (D) Correlation heatmap showing the Pearson correlation ($r$) values between UCLA4 MER51B-GFP$^+$ and MER51B-GFP$^+$LTR5_Hs-RFP$^+$ RNA-seq samples. The scale bar represents the range of the correlation coefficients ($r$) displayed. (E) Mean intensity of MER51B-GFP and LTR5_Hs-RFP on gated MER51B-GFP$^+$LTR5_Hs-RFP$^+$ cells in primed versus 4CL condition. Each data point represents an independent biological replicate ($n = 3$). Statistical significance was determined by unpaired two-tailed Student's $t$-test, and exact $p$ values are represented in the figure. Data were presented as mean ± standard deviation. (F) qRT-PCR for the indicated genes in primed MER51B-GFP$^+$LTR5_Hs-RFP$^+$ UCLA5 cells. Each data point represents an independent biological replicate ($n = 3$). Statistical significance was determined by unpaired two-tailed Student's $t$-test, and exact $p$ values are represented in the figure. Data were presented as mean ± standard deviation. Relative gene expression was normalized to *ACTB*. (G) Representative flow cytometry plots showing basal H3Y-mCherry expression in primed UCLA4 cells. (H) Flow cytometric quantification for the percentage of γ-H2AX$^+$ cells in gated MER51B-GFP$^+$LTR5_Hs-RFP$^+$ versus MER51B-GFP$^+$ UCLA4 cells cultured in 4CL. Each data point represents an independent biological replicate ($n = 3$). Statistical significance was determined by unpaired two-tailed Student's $t$-test, and exact $p$ values are represented in the figure. $P$ values $>0.05$ are non-significant. Data were presented as mean ± standard deviation. (I) Representative flow cytometry plots showing primed MER51B-GFP$^+$/LTR5_Hs-RFP$^+$ cells 40 h after washing off DMSO or cisplatin treatment (treatment length: 10 h). (J) Representative flow cytometry plots showing primed MER51B-GFP/LTR5_Hs-RFP cells 12 h after washing off DMSO or hydroxyurea treatment (treatment length: 12 h). (K) qRT-PCR for the indicated genes in bulk-primed cells treated with doxorubicin for 8 h. Each data point represents an independent biological replicate ($n = 3$). Statistical significance was determined by unpaired two-tailed Student's $t$-test, and exact $p$ values are represented in the figure. Data were presented as mean ± standard deviation. Relative gene expression was normalized to *ACTB*.

**A**

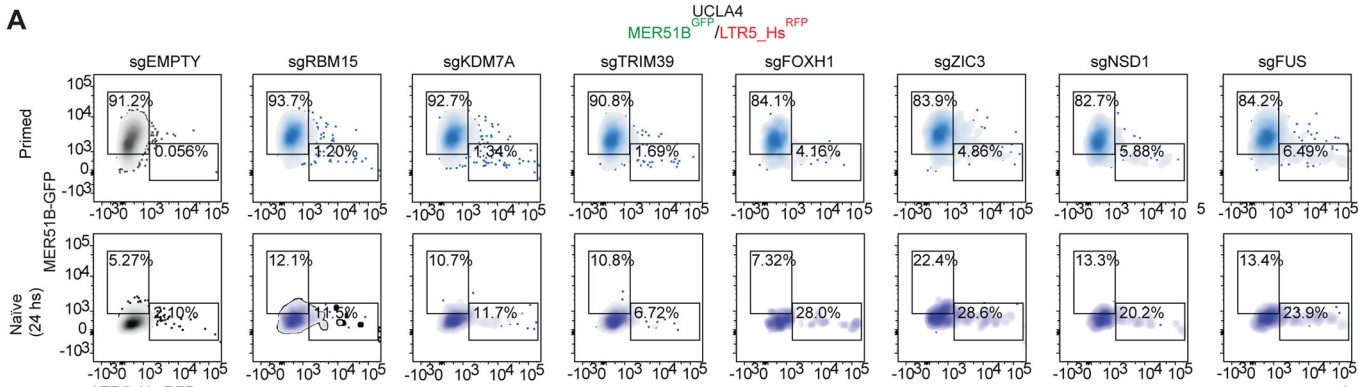

**B**

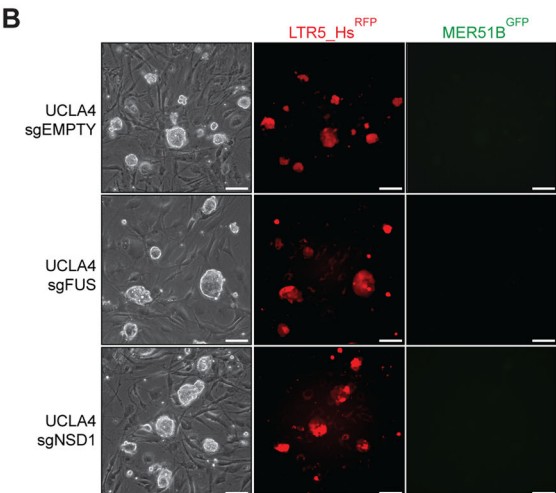

**Figure EV4. Use of MER51B and LTR5_Hs dual reporter system to study regulators of stem cell potency.**

(**A**) Representative flow cytometry plots showing changes in MER51B-GFP and LTR5_Hs-RFP expression after knockout of the indicated genes, in primed medium or naïve culture conditions for 24 h. (**B**) Representative phase contrast and fluorescence microscopy images of established naïve cell cultures (P3) of UCLA4 MER51B-GFP/LTR5_Hs-RFP hESC after knockout of FUS and NSD1. Scale bar: 300 µm.

