## [Peer Review File · EMBO Reports]

Transposable element activity captures human pluripotent cell states

Florencia Levin-Ferreyra, Srikanth Kodali, Yingzhi Cui, Alison Pashos, Patrizia Pessina, Justin Brumbaugh, and Bruno Di Stefano

Corresponding author(s): Bruno Di Stefano (bruno.distefano@bcm.edu)

Review Timeline:

Submission Date:	4th Aug 24
Editorial Decision:	13th Sep 24
Revision Received:	1st Nov 24
Editorial Decision:	19th Nov 24
Revision Received:	21st Nov 24
Accepted:	25th Nov 24

Editor: Achim Breiling

Transaction Report:

Dear Dr. Di Stefano

Thank you for the submission of your manuscript to EMBO reports. I have now received the reports from the three referees that were asked to evaluate your study, which can be found at the end of this email.

As you will see, the referees find the study interesting. However, they have several comments, concerns, and suggestions that need to be addressed to allow publication of the study in EMBO reports. As the reports are below, and all the concerns need to be addressed, I will not detail them further here.

Acceptance of your manuscript will depend on a positive outcome of a second round of review. It is EMBO reports policy to allow a single round of revision only and acceptance of the manuscript will therefore depend on the completeness of your responses included in the next, final version of the manuscript.

1) a .docx formatted version of the final manuscript text (including legends for main figures, EV figures and tables), but without the figures included. Figure legends should be compiled at the end of the manuscript text.

2) individual production quality figure files as .eps, .tif, .jpg (one file per figure), of main figures and EV figures. Please upload these as separate, individual files upon re-submission.

4) a complete author checklist, which you can download from our author guidelines

(<https://www.embopress.org/page/journal/14693178/authorguide>). Please insert page numbers in the checklist to indicate where the requested information can be found in the manuscript. The completed author checklist will also be part of the RPF.

5) that primary datasets produced in this study (e.g. RNA-seq, ChIP-seq, structural and array data) are deposited in an appropriate public database. If no primary datasets have been deposited, please also state this in a dedicated section (e.g. 'No primary datasets have been generated and deposited'), see below.

The accession numbers and database should be listed in a formal "Data Availability" section (placed after Materials & Methods) that follows the model below. This is now mandatory (like the COI statement). Please note that the Data Availability Section is restricted to new primary data that are part of this study. This section is mandatory. As indicated above, if no primary datasets have been deposited, please state this in this section

Data availability

8) Regarding data quantification and statistics, please make sure that the number "n" for how many independent experiments were performed, their nature (biological versus technical replicates), the bars and error bars (e.g. SEM, SD) and the test used to calculate p-values is indicated in the respective figure legends (also for EV figures and all those in an Appendix). Please also check that all the p-values are explained in the legend, and that these fit to those shown in the figure. Please provide statistical testing where applicable. Please avoid the phrase 'independent experiment', but clearly state if these were biological or technical replicates. Please also indicate (e.g. with n.s.) if testing was performed, but the differences are not significant. In case n=2, please show the data as separate datapoints without error bars and statistics. See also: <http://www.embopress.org/page/journal/14693178/authorguide#statisticalanalysis>

9) Please add scale bars of similar style and thickness to microscopic images, using clearly visible black or white bars (depending on the background). Please place these in the lower right corner of the images themselves. Please do not write on or near the bars in the image but define the size in the respective figure legend.

10) Please also note our reference format:

12) We now use CRediT to specify the contributions of each author in the journal submission system. CRediT replaces the author contribution section. Please use the free text box to provide more detailed descriptions and do NOT provide your final manuscript text file with an author contributions section. See also our guide to authors: <https://www.embopress.org/page/journal/14693178/authorguide#authorshipguidelines>

13) All Materials and Methods need to be described in the main text using our 'Structured Methods' format, which is required for all research articles. According to this format, the Materials and Methods section should include a Reagents and Tools Table (listing key reagents, experimental models, software, and relevant equipment and including their sources and relevant

identifiers), uploaded as separate file, followed by a Methods and Protocols section in which we encourage the authors to describe their methods using a step-by-step protocol format with bullet points, to facilitate the adoption of the methodologies across labs. More information on how to adhere to this format as well as downloadable templates (.doc) for the Reagents and Tools Table can be found in our author guidelines (section 'Structured Methods'):

14) Please add up to 5 keywords to the manuscript and order the manuscript sections like this, using these names:

Title page - Abstract - Keywords - Introduction - Results & Discussion - Methods - Data availability section - Acknowledgements (including funding information) - Disclosure and Competing Interests Statement - References - Figure legends - Expanded View Figure legends

I look forward to seeing a revised form of your manuscript when it is ready.

Yours sincerely,

Referee #1:

In this manuscript, the authors developed a transposon-based dual reporter system to track the dynamics of pluripotent states. With this reporter system, a rare, metastable cell population in primed hESCs, which expresses low levels of genes associated with early pre-implantation embryo development and triggered by DNA damage, was identified. In addition, the authors validated the role of FUS and NSD1 in the primed-to-naïve conversion in the reporter hESCs. The development of a reporter system for various pluripotent states is of interest for the field of stem cell research. Yet, the discoveries based on the reporter system lack novelty or biological significance.

Major concerns:

1. What kind of cells are MER51B-GFP+ and LTR5_Hs-RFP+ in different experimental setting? The double positive cell populations seem to be different in 4CL-induced hPSCs and primed hPSCs. It argues against the conclusion that the reporter system faithfully and reproducibly track the dynamics of hPSC cell state transitions.
2. Many experiments have been performed to validate that MER51B-GFP is a good reporter for primed pluripotency. Yet, whether LTR5_Hs-RFP could allow us to track the transition from naïve pluripotency, was not sufficiently demonstrated. Similar experiments should be performed with naïve hESCs.
3. What is the biological function of GFP+/RFP+ population in primed hPSCs?
4. The evidence for that DNA damage leads to the GFP+/RFP+ population in primed hPSCs is weak. What percentage of cells are associated with DNA damage upon cisplatin and hydroxyurea treatment? Why do only around 2% or 8% of cells become GFP+/RFP+, when treated with cisplatin or hydroxyurea?
5. "The LTR5_Hs and MER51B reporter systems facilitate discovery of novel key regulators of cell potency", and "MER51B and LTR5_Hs activity uncover new regulators of stem cell potency". These sentences are overstated. The authors only utilized the reporter system to validate pluripotency regulators identified in other studies.

Minor concerns:

1. Line 149, "human hPSCs" is redundant. Please specify naïve or primed hPSCs.
2. Some experimental procedures were not clearly described, such as Fig. 4A.

Referee #2:

The manuscript by the Di Stefano lab describes a new reporter system to monitor pluripotency transitions.

The manuscript is very well written and easy to read, the figures are very clear.

More importantly, the number of independent assays used to test the reporter system is really remarkable and show conclusively that it is a good system.

I have only some two minor points to should be addressed:

1. in the primed to naive conversion (Figure 2A and EV2A) the authors showed that their reporter system displayed dynamics similar to the CD90/CD75 system.

I think it would be informative to show what is the fraction of GFP-positive cells that are also positive for CD90, and the fraction of RFP-positive cells that are also CD75 positive.

2. Results in figure 4, about resetting to naive pluripotency suggest that the reporter system can be used to find functional regulators of pluripotency transition. However, it would be more convincing to show that the RFP+ cells, obtained for example with sgFUS, can be sorted by FACS and expanded for multiple passages under naive conditions. Such experiment would indicate the correct and stable acquisition of naive identity.

Referee #3:

The paper entitled 'Transposable element activity captures human pluripotent states' identified naïve and primed specific TE activity, which represents the two different pluripotent state. Authors assessed their reporter system in reprogramming back to naïve as well as differentiating further down to the somatic lineage. They also characterised rare population exists in the prime culture. TE activity based reporter system to distinguish human naïve and primed pluripotency can be a useful system to the stem cell community. The manuscript is well written and the data presented here is mostly convincing enough to support authors' claim. Therefore, I only have minor questions and comments to improve this manuscript.

1. How authors narrow down to identify LTR5_Hs as naïve specific and MER51B as primed specific TEs? Fig. EV1A shows more candidates TEs, it is unclear and curious to know why these two TEs were chosen for further analyses.
2. Fig.1 F and EV1F. Did authors sort GFP+ or RFP+ population to perform RNA-seq analysis? This was not clear from main text or method section.
3. This reviewer is interested in seeing the kinetics of CD90 expression pattern with MER51B-GFP, as well as CD75 emergence with LTR5_Hs_RFP in Fig EV2A.
4. How are these TEs expressed in human embryos? It is very useful if authors provide a figure showing TE activities during early development.
5. Fig. 2C-D. 4CL conversion induced double positive, RFP single and GFP single positive and double negative fractions. Are double negative cells under 4CL pluripotent? Why authors excluded the RFP single and double negative from the analysis?
6. The cell type labelling of Fig 2J is misplaced.
7. Fig. 3D ATAC-seq analysis. Can authors really distinguish the transgene derived fragments from endogenous TE elements? Some amounts could be topped up from the transgene.
8. Fig. 3F should include 4CL subpopulations used in Fig.2. Or at least some analysis needed to show that double positive populations in primed and 4CL conditions are different in the supplementary figure.
9. Fig. 3I, K. It is unclear the 8C cells or 8CLCs are the same samples and these are 4CL samples generated in this study or the data published in Mazid et al 2022.
10. Fig. 4D. ZFP42 is not a good marker for the naïve state. KLF4 or -17 is more appropriate here.
11. In the naïve state, there seems to be another rare population of double positive exits. Are they also up-regulate DNA damage, ZGA and 8-cell genes?
12. Most of the figures are well prepared and presented but some figure legends have minimum explanations and there is not more information than shown in the figure itself (see comments 2 and 8 above, and for example, in qPCR data, no information is provided what they used for normalisation). Please check through again and add more information where necessary to understand each figure.

Response to reviewers' comments on EMBO Reports submission EMBOR-2024-60136V1. "Transposable element activity captures human pluripotent cell states."

We thank the editor and reviewers for their valuable comments, which allowed us to substantially improve our manuscript. To reinforce our original conclusions and broaden the scope of our study, we have incorporated extensive new data. The main points are summarized below:

- 1. Characterization of transcriptional identity of cells cultured under 4CL conditions:** We have conducted comprehensive transcriptional characterization of distinct cell populations identified by our dual reporter system under 4CL culture conditions. Our analysis reveals that LTR5_Hs-RFP⁺ cells express the highest levels of naïve pluripotency markers, while MER51B-GFP⁺ cells predominantly express primed state markers. These findings validate our dual reporter system as an effective tool for distinguishing naïve and primed pluripotent states in heterogeneous cultures. Furthermore, we demonstrate that GFP⁺/RFP⁺ double-positive cells in 4CL-medium express pre-implantation transcripts at levels comparable to TPRX1⁺ 8C-like cells. See pages 5 and 7, and **Figures 2E, 3G, and EV2D, E.**
- 2. Extended characterization of transposon activity in pre-implantation embryos:** To validate our observation that LTR5_Hs and MER51B exhibit mutually exclusive activity in pre-implantation embryonic cells, we analyzed published scRNA-seq and scATAC-seq datasets from human embryos across various pre-implantation developmental stages. Our analyses demonstrate that LTR5_Hs elements show both expression and accessibility during morula and early blastocyst stages, aligning with their high expression in naïve and 8C-like cells *in vitro*. Conversely, MER51B elements remain transcriptionally silent and lowly accessible during these stages, corroborating our *in vitro* findings that their activation corresponds to the transition toward primed pluripotency, specifically in cells that resemble the post-implantation epiblast. Please see page 4 and **Figures EV1E, F.**
- 3. Evidence that DNA damage triggers a metastable cell population marked by LTR5_Hs and MER51B activity in primed conditions:** To strengthen the connection between DNA damage and activation of our dual reporter system, we expanded our analysis using multiple DNA-damaging agents, examining the relationship between GFP⁺/RFP⁺ cell induction in primed cells and γ -H2AX activation, a canonical DNA damage marker. Our results show that, upon treatment with cisplatin and hydroxyurea, the emergence of GFP⁺/RFP⁺ cells is closely correlated with that of γ -H2AX-positive cells, reinforcing the connection between double-positive populations and the DNA damage response in primed cells. See pages 7-8 and **Figures 3N-Q and EV3I, J.**
- 4. Extension of key observation during re-priming experiments:** To further demonstrate that our reporter system reliably tracks the dynamic transitions between naïve and primed pluripotent states, we monitored MER51B and LTR5_Hs activity during the conversion of naïve hPSCs to primed states using established re-priming protocols. Our results reveal that induction of primed pluripotency leads to rapid suppression of LTR5_Hs expression concurrent with MER51B activation, resulting in an RFP⁻/GFP⁺ primed cell population within two weeks. These data highlight the robustness of our reporter system in tracking

the transitions between pluripotent states. These findings are detailed on page 5 and in **Figures 2F and EV2F**.

Reviewer#1:

Remarks to the Author:

In this manuscript, the authors developed a transposon-based dual reporter system to track the dynamics of pluripotent states. With this reporter system, a rare, metastable cell population in primed hESCs, which expresses low levels of genes associated with early pre-implantation embryo development and triggered by DNA damage, was identified. In addition, the authors validated the role of FUS and NSD1 in the primed-to-naïve conversion in the reporter hESCs. The development of a reporter system for various pluripotent states is of interest for the field of stem cell research. Yet, the discoveries based on the reporter system lack novelty or biological significance.

Response: We thank the reviewer for recognizing the utility of our reporter system for the stem cell field. We would like to emphasize key, novel findings reported in our study, as detailed in the responses below. Our work is the first to characterize the transposable elements MER51B and LTR5_Hs as sensitive, accurate indicators of primed and naïve human pluripotent stem cell states. Of note, MER51B activity surpasses the accuracy of currently available reporter systems that are based on the expression of pluripotency factors such as NANOG and OCT4, which are expressed in both naïve and primed pluripotent cells. Moreover, using this system, we identified a rare, metastable cell population within primed cells that expresses genes associated with totipotency and DNA damage. This finding has relevance for understanding how DNA damage and the activity of transposable elements are related within a homogeneous cell population. Most notably, our data strongly caution against using p53 activators or DNA damage-inducing agents in cell culture media to induce or culture 8C-like cells, as these may lead to superficial transcriptional activation of ZGA genes without the concomitant acquisition of 8C-like cell potency. We believe that these points are relevant to the field and will be of widespread interest.

1. What kind of cells are MER51B-GFP+ and LTR5_Hs-RFP+ in different experimental setting? The double positive cell populations seem to be different in 4CL-induced hPSCs and primed hPSCs. It argues against the conclusion that the reporter system faithfully and reproducibly track the dynamics of hPSC cell state transitions.

Response: We thank the reviewer for this important observation, which has led us to perform additional analyses that further validate our reporter system's reliability. We have now conducted detailed comparisons of GFP and RFP expression intensities in double-positive cells under both primed and 4CL conditions. Our flow cytometry analyses demonstrate distinct reporter expression patterns that correlate with cellular identity: cells in 4CL conditions exhibit significantly higher LTR5_Hs-RFP and lower MER51B-GFP expression compared to those in primed conditions (**Fig. EV3E**). Importantly, these differences in reporter intensity align with functional cellular states. Double-positive cells cultured in 4CL conditions express 8C genes at levels comparable to bona fide 8C-like cells, while those in primed conditions do not (**Figs. 3J and EV2D**). Critically, unlike double-positive cells in primed conditions, GFP⁺RFP⁺ cells in 4CL show no elevation in DNA

damage markers (**Fig EV3H**). Rather than contradicting our system's reliability, these findings demonstrate that our dual reporter can capture both qualitative and quantitative aspects of pluripotent cell state transitions. The system not only identifies distinct cell populations but also provides insights into their developmental trajectory through the relative intensities of reporter expression.

2. Many experiments have been performed to validate that MER51B-GFP is a good reporter for primed pluripotency. Yet, whether LTR5_Hs-RFP could allow us to track the transition from naïve pluripotency, was not sufficiently demonstrated. Similar experiments should be performed with naïve hESCs.

Response: We agree that our original submission would benefit from a more thorough characterization of the dynamics of LTR5_Hs and MER51B during the transition from naïve to primed pluripotency. To address this important point, we have added additional experiments and analyses to expand the scope and depth of our study. Specifically, we now demonstrate that the induction of primed pluripotency from naïve cells using established protocols (Rostovskaya *et al.*, Development 2019) leads to the rapid extinction of LTR5_Hs expression and the activation of the MER51B reporter, with cells becoming predominantly RFP⁻/GFP⁺ within two weeks. We further demonstrate that reversion of reprimed cells to naïve pluripotency is, conversely, accompanied by reactivation of LTR5_Hs and loss of MER51B, highlighting the reliability of our reporter system in tracking transitions between pluripotent states. The new data are presented in **Fig. 2F** and **EV2F**.

3. What is the biological function of GFP⁺/RFP⁺ population in primed hPSCs?

Response: We thank the reviewer for this important question about the biological significance of the GFP⁺/RFP⁺ population in primed hPSCs. Our data reveal that this rare, metastable population represents a distinct cellular state that emerges in response to DNA damage, similar to recently described populations in cancer cells (Smith *et al.*, Cell Reports 2023). While these cells express pre-implantation and ZGA-associated genes, their overall transcriptional and chromatin profiles remain largely similar to conventional primed cells, indicating they are not genuine totipotent-like cells. This finding has immediate practical implications, as it challenges previous reports suggesting DNA damage inducers can generate bona fide 8C-like cells (Yu, *et al.*, Cell Reports, 2022). More broadly, these cells provide a valuable experimental system for investigating the mechanistic links between DNA damage, transcriptional regulation, and transposable element activity. Our observations suggest the existence of regulatory mechanisms that prevent cells experiencing DNA damage from fully activating ZGA genes to levels required for stable 8C-like cell conversion. This insight positions our reporter system as a powerful tool for identifying transcriptional and post-transcriptional barriers to totipotency. Future investigations will address whether these DNA-damaged cells are ultimately eliminated from cultures and explore whether transposable element activation serves as an adaptive DNA damage response signal or represents a maladaptive consequence in both stem cells and cancer. We have revised the manuscript to better contextualize these interpretations and their significance.

4. *The evidence for that DNA damage leads to the GFP+/RFP+ population in primed hPSCs is weak. What percentage of cells are associated with DNA damage upon cisplatin and hydroxyurea treatment? Why do only around 2% or 8% of cells become GFP+/RFP+, when treated with cisplatin or hydroxyurea?*

Response: To address this point, we have optimized the treatment protocols for cisplatin and hydroxyurea, with details reported in the updated methods section. Given the sensitivity of hPSCs to these cytotoxic agents, we treat the cells for only a few hours, followed by washout and culture in stem cell medium. While this results in low percentages of cells exhibiting DNA damage, it also minimizes confounding effects related to cell death and the presence of apoptotic cells in the culture. Moreover, we have broadened our characterization of cisplatin and hydroxyurea treatment by assessing the induction of GFP+/RFP+ cells alongside the concomitant activation of γ -H2AX as a marker for DNA damage. As illustrated in **Fig. 3P**, our findings consistently reveal that the percentage of GFP+/RFP+ cells that emerge upon treatment with either cisplatin (~6%) or hydroxyurea (~8%) closely correlates with the appearance of γ -H2AX-positive cells (~6 and ~9%, respectively). This further strengthens the connection between the double-positive cell populations and the DNA damage response.

5. *"The LTR5_Hs and MER51B reporter systems facilitate discovery of novel key regulators of cell potency", and "MER51B and LTR5_Hs activity uncover new regulators of stem cell potency". These sentences are overstated. The authors only utilized the reporter system to validate pluripotency regulators identified in other studies.*

Response: As noted by the reviewer, we analyzed previous loss-of-function screens conducted in human PSCs to identify regulators that may control cell potency. It is important to emphasize that the vast majority of reported hits—including those analyzed in our manuscript—were unvalidated and unexplored in the original studies. Nonetheless, we also acknowledge that future studies will be necessary to investigate the mechanisms underlying the roles of FUS and NSD1 in pluripotent stem cells. We have thus de-emphasized the text to reflect the reviewer's comments and rephrased it to ensure greater accuracy and avoid overstatement.

Minor concerns:

1. *Line 149, "human hPSCs" is redundant. Please specify naïve or primed hPSCs.*
2. *Some experimental procedures were not clearly described, such as Fig. 4A.*

Response: We apologize for the lack of clarity and have updated the text and methods accordingly.

Reviewer#2:

Remarks to the Author:

The manuscript by the Di Stefano lab describes a new reporter system to monitor pluripotency transitions. The manuscript is very well written and easy to read, the figures are very clear. More importantly, the number of independent assays used to test the reporter system is really remarkable and show conclusively that it is a good system.

Response: We would like to thank the reviewer for their interest in our study and for the positive feedback regarding our manuscript.

Minor points:

1. *in the primed to naive conversion (Figure 2A and EV2A) the authors showed that their reporter system displayed dynamics similar to the CD90/CD75 system. I think it would be informative to show what is the fraction of GFP-positive cells that are also positive for CD90, and the fraction of RFP-positive cells that are also CD75 positive.*

Response: To address this point, we have included additional experimental data and revised our figures and text. Specifically, we examined the kinetics of GFP and RFP expression in conjunction with the expression patterns of THY-1 (CD90) and CD75. Our results indicate that MER51B and LTR5_Hs serve as reliable indicators of THY-1 and CD75 expression, and that their dynamic activity during the transition from primed to naïve reprogramming mirrors the patterns observed for THY-1 and CD75. We have modified the figures to highlight the fraction of GFP⁺ cells that are also positive for CD90 and the fraction of RFP⁺ cells that are also CD75 positive (**Fig. 2B and EV2B**).

2. *Results in figure 4, about resetting to naive pluripotency suggest that the reporter system can be used to find functional regulators of pluripotency transition. However, it would be more convincing to show that the RFP+ cells, obtained for example with sgFUS, can be sorted by FACS and expanded for multiple passages under naive conditions. Such experiment would indicate the correct and stable acquisition of naive identity.*

Response: This is an excellent point. We have now included data showing that we can efficiently establish naïve cell cultures following the expression of sgFUS and sgNSD1. Our results show that, for several passages, sgFUS and sgNSD1 naïve cells maintain LTR5_Hs expression and exhibit dome-shaped colony morphology, indicating stable acquisition of naïve identity (**Fig. EV4B**).

Reviewer#3:

Remarks to the Author:

The paper entitled 'Transposable element activity captures human pluripotent states' identified naïve and primed specific TE activity, which represents the two different pluripotent state. Authors assessed their reporter system in reprogramming back to naïve as well as differentiating further down to the somatic lineage. They also characterized rare population exists in the prime culture. TE activity-based reporter system to distinguish human naïve and primed pluripotency can be a useful system to the stem cell community. The manuscript is well written, and the data presented here is mostly convincing enough to support authors' claim.

Response: We thank the reviewer for their enthusiasm and interest in our study.

Minor points:

1. How authors narrow down to identify LTR5_Hs as naïve specific and MER51B as primed specific TEs? Fig. EV1A shows more candidates TEs, it is unclear and curious to know why these two TEs were chosen for further analyses.

Response: We apologize for not clearly conveying the selection of LTR5_Hs and MER51B. Our transcriptomic data show that these two transposable elements are the most differentially expressed between primed and naïve hPSCs. Their distinct expression patterns make them robust markers for distinguishing between these pluripotent states. We have also revised the figures, and included a new bioinformatic analysis on **Fig. 1A**, to more clearly highlight this point.

2. Fig.1 F and EV1F. Did authors sort GFP+ or RFP+ population to perform RNA-seq analysis? This was not clear from main text or method section.

Response: We apologize for the lack of clarity regarding the RNA-seq analysis. We have now explicitly clarified in both the figure legend and the main text that the data presented were derived from GFP⁺ and RFP⁺ sorted cell populations. This information has also been added to the methods section for completeness.

3. This reviewer is interested in seeing the kinetics of CD90 expression pattern with MER51B-GFP, as well as CD75 emergence with LTR5_Hs_RFP in Fig EV2A.

Response: To address this point, we have conducted additional experiments and revised our figures and text (**Fig. 2B**). Specifically, we assessed the kinetics of GFP and RFP expression alongside THY-1 (CD90) and CD75 expression patterns. Our results show that MER51B and LTR5_Hs activity during the transition from primed to naïve reprogramming mirrors the expression patterns of THY-1 and CD75.

4. How are these TEs expressed in human embryos? It is very useful if authors provide a figure showing TE activities during early development.

Response: Thank you for raising this important point. To address it, we analyzed scRNA-seq and scATAC-seq data from pre-implantation human embryos (Xue *et al.*, Nature, 2013 and Liu *et al.*, Nat. Comm., 2019). Our analysis shows that LTR5_Hs transcripts are detectable during the morula and early blastocyst stages, consistent with their high expression in naïve pluripotent stem cells and 4CL-cultured cells *in vitro*. Additionally, we observed increased chromatin accessibility at LTR5_Hs genomic regions during the morula-to-early-blastocyst transition. In contrast, MER51B elements are lowly expressed and accessible during pre-implantation development, which supports our *in vitro* findings that their expression is activated during the transition from naïve to primed pluripotency, specifically in primed cells resembling post-implantation epiblast cells. Unfortunately, robust datasets covering post-implantation human embryo development are still lacking. The few available datasets from cultured embryos or embryo-like structures lack sufficient sequencing coverage to reliably analyze transposable elements with current tools. We have included these analyses in **Fig. EV1E, F**.

5. Fig. 2C-D. 4CL conversion induced double positive, RFP single and GFP single positive and double negative fractions. Are double negative cells under 4CL pluripotent? Why authors excluded the RFP single and double negative from the analysis?

Response: We intentionally excluded the RFP single-positive and double-negative cells from the original RNA-seq analysis because our primary focus was the comparison between double-positive populations across culture conditions. However, to address the reviewer's question, we have now isolated RFP⁺, GFP⁺, double-negative (RFP⁻/GFP⁻), and double-positive (RFP⁺/GFP⁺) cells and analyzed the expression of transcripts related to totipotency, primed and naïve pluripotency using qRT-PCR. Our analysis shows that RFP⁺ cells express the highest levels of naïve pluripotency genes, while GFP⁺ cells express primed markers, reinforcing our conclusion that MER51B and LTR5_HS activity effectively mark the naïve and primed pluripotent states in mixed cultures. Interestingly, the double-negative population shows elevated naïve pluripotency marker expression compared to primed cells, suggesting these cells may represent an intermediate state transitioning toward naïve pluripotency. Finally, as expected from our original RNA-seq analysis, the double-positive (RFP⁺/GFP⁺) population expresses the highest levels of 8C-related genes transcripts among the cell populations tested. These new data are now presented in **Fig. EV2E**.

6. The cell type labelling of Fig 2J is misplaced.

Response: We thank the reviewer for bringing this to our attention and have made the relevant correction.

7. Fig. 3D ATAC-seq analysis. Can authors really distinguish the transgene derived fragments from endogenous TE elements? Some amounts could be topped up from the transgene.

Response: This is an excellent point. To address the concern about whether the ATAC-seq analysis was influenced by the presence of the transgenes, we reanalyzed the data, filtering out sequencing reads derived from the piggyBac constructs. Our new results, now presented in **Fig. 3E**, confirm our original findings. Specifically, primed cells marked by the concomitant expression of LTR5_Hs and MER51B still show increased accessibility at both the LTR5_Hs and MER51B genomic regions. Therefore, the presence of the transgene does not affect the overall conclusions.

8. Fig. 3F should include 4CL subpopulations used in Fig.2. Or at least some analysis needed to show that double positive populations in primed and 4CL conditions are different in the supplementary figure.

Response: We apologize for the oversight and have corrected the labeling in the relevant panels. In addition, our PCA analysis, now presented in **Fig. 3G**, demonstrates that the double-positive cells from primed and 4CL cultures are significantly different at the transcriptional level.

9. Fig. 3I, K. It is unclear the 8C cells or 8CLCs are the same samples and these are 4CL samples generated in this study or the data published in Mazid et al 2022.

Response: In **Fig. 3J**, we compared the gene expression data of our GFP⁺ and GFP⁺/RFP⁺ primed cells with published 8C-like cells, using gene signatures from both primed and naïve states (Messmer *et al.*, Cell Reports 2019), as well as ZGA-related transcripts (Taubenschmid-Stowers *et al.*, Cell Stem Cell 2022). The 8C-like cell data shown in **Fig. 3L** were generated in our lab using the TPRX1-GFP reporter, as described in Mazid *et al.*, Nature, 2022. We have updated the text and methods section to clarify this distinction.

10. Fig. 4D. ZFP42 is not a good marker for the naïve state. KLF4 or -17 is more appropriate here.

Response: We appreciate the suggestion and have now included qRT-PCR data for *KLF17* in the sgFUS and sgNSD1 samples (**Fig. 4D**). Our results demonstrate that the knockout of these factors accelerates the conversion of primed cells to a naïve pluripotency state.

11. In the naïve state, there seems to be another rare population of double positive exits. Are they also up-regulate DNA damage, ZGA and 8-cell genes?

Response: Thank you for raising this point. We have conducted an extensive analysis of our reporter system in cells cultured under naïve conditions. Our data reveal the presence of an extremely rare population of low-intensity GFP⁺RFP⁺ cells (<0.05%). To further investigate, we sorted these cells and performed qRT-PCR analysis. Our results show that double-positive cells from 5iLAF-cultured naïve conditions do not activate ZGA or DNA damage-related genes (please see the figure below (**Fig. R1**)). Since this population is rare and inconsistent across biological replicates, we have opted not to include these data in the main text.

Figure R1. qRT-PCR for the indicated genes in sorted 5i/LAF naïve MER51B-GFP⁺LTR5_Hs-RFP⁺ versus bulk UCLA4 naïve cells (n=3, mean ± s.d., unpaired two-tailed Student's t-test).

12. Most of the figures are well prepared and presented but some figure legends have minimum explanations and there is not more information than shown in the figure itself (see comments 2 and 8 above, and for example, in qPCR data, no information is provided what they used for

normalisation). Please check through again and add more information where necessary to understand each figure.

Response: We thank the reviewer for bringing this to our attention and have made the corrections in the text.

Dear Dr. Di Stefano,

Thank you for the submission of your revised manuscript to our editorial offices. I have now received the reports from the referees that I asked to re-evaluate the study, you will find below. As you will see, all three referees now support publication of the study in EMBO reports. Referee #3 has a final comment I ask you to address in a final revised manuscript.

Moreover, I have these further editorial requests:

- Please reduce the number of keywords to 5 and order the manuscript sections using (only) these names: Abstract - Keywords - Introduction - Results & Discussion - Methods - Data availability section - Acknowledgements (including funding information) - Disclosure and Competing Interests Statement - References - Figure legends - Expanded View Figure legends
- Please add scale bars of similar style and thickness to all microscopic images, using clearly visible black or white bars (depending on the background). Please place these in the lower right corner of the images themselves. Please do not write on or near the bars in the image but define the size in the respective figure legend. Presently, several scale bars are too thin or can hardly be seen (e.g. in Fig. EV2G). Please check.
- Please check again that the number "n" for how many independent experiments were performed, their nature (biological versus technical replicates), the bars and error bars (e.g. SEM, SD) and the test used to calculate p-values is indicated in the respective figure legends (main, EV and Appendix figures). Please also check that all the p-values are explained in the legend, and that these fit to those shown in the figure. Please provide statistical testing where applicable. Please avoid the phrase 'independent experiment', but clearly state if these were biological or technical replicates. Please also indicate (e.g. with n.s.) if testing was performed, but the differences are not significant. In case $n=2$, please show the data as separate datapoints without error bars and statistics. See also:
<http://www.embopress.org/page/journal/14693178/authorguide#statisticalanalysis>
- If $n < 5$, please show single datapoints for diagrams. Moreover:
 - Please indicate the statistical test used for data analysis in the legends of figures 1a; 3h; 4b.
 - Please note that the box plots need to be defined in terms of minima, maxima, centre, bounds of box and whiskers, and percentile in the legends of figures 3j; EV 2d.
 - Please note that information related to n is missing in the legends of figures 3j; EV 1e; EV 2d; Ev 3e.
 - Please note that the error bars are not defined in the legends of figures 2d; 3b; EV 3e.
- Please add the primer information provided in the Methods section to the Reagents and tools table and add a callout (e.g. 'see reagents and tools table'). Please also remove the instructions and the empty template table from the reagents and tools table.
- Please make sure that all the funding information is also entered into the online submission system and that it is complete and similar to the one in the acknowledgement section of the manuscript text file. It seems that the Cytometry and Cell Sorting Core at Baylor College of Medicine with funding from the CPRIT Core Facility Support Award (CPRIT-RP180672) and the NIH grant (CA125123 and RR024574) are missing from the submission system. Please check.
- Thank you for providing the requested source data. Please upload this as one folder per figure (with all files for one figure in one folder and ZIPed together).

In addition, I would need from you uploaded separately:

Referee #1:

The authors have addressed my concerns appropriately. And I have no more comments.

Referee #2:

The authors addressed in a satisfactory way all comments raised by reviewers.
The authors generated an extensive set of new data and made their conclusions more solid, thus I fully support publication of the revised manuscript.

Referee #3:

In this revision, authors sufficiently responded to all my concerns. The revised version shows the difference of double positive cells between 4CL and primed culture much clearer. I found the manuscript is suitable for the publication at EMBO reports.

Please check Fig. 3I before the publication. This figure highlighted differentially expressed genes. I believed the listed genes here are the part of them which are enriched in double positive fraction, but not all the genes. The bottom line is not properly placed, and it might be better show examples of GFP+ genes to clarify.

Response to editorial comments on EMBO Reports EMBOR-2024-60136V2: " Transposable element activity captures human pluripotent cell states."

- Please reduce the number of keywords to 5 and order the manuscript sections using (only) these names:

Abstract - Keywords - Introduction - Results & Discussion - Methods - Data availability section - Acknowledgements (including funding information) - Disclosure and Competing Interests Statement - References - Figure legends - Expanded View Figure legends

Response: We have reorganized the manuscript sections as requested and reduced the keywords to five.

- Please add scale bars of similar style and thickness to all microscopic images, using clearly visible black or white bars (depending on the background). Please place these in the lower right corner of the images themselves. Please do not write on or near the bars in the image but define the size in the respective figure legend. Presently, several scale bars are too thin or can hardly be seen (e.g. in Fig. EV2G). Please check.

Comments:

Response: We have revised all microscopic images to include clearly visible scale bars of consistent style and thickness. These are now positioned in the lower right corner of each image, with their sizes defined in the corresponding figure legends.

- Please check again that the number "n" for how many independent experiments were performed, their nature (biological versus technical replicates), the bars and error bars (e.g. SEM, SD) and the test used to calculate p-values is indicated in the respective figure legends (main, EV and Appendix figures). Please also check that all the p-values are explained in the legend, and that these fit to those shown in the figure. Please provide statistical testing where applicable. Please avoid the phrase 'independent experiment', but clearly state if these were biological or technical replicates. Please also indicate (e.g. with n.s.) if testing was performed, but the differences are not significant. In case n=2, please show the data as separate datapoints without error bars and statistics.

Response: We apologize for the earlier inaccuracies. All figure legends (main and EV) have been updated to clearly define the number of replicates ("n"), their nature (biological or technical), error bars (e.g., SEM, SD), and the statistical tests used to calculate p-values. Exact p-values are indicated for each panel, and we have defined $p > 0.05$ as non-significant.

If $n < 5$, please show single datapoints for diagrams. Moreover:

- Please indicate the statistical test used for data analysis in the legends of figures 1a; 3h; 4b.

- Please note that the box plots need to be defined in terms of minima, maxima, centre, bounds of box and whiskers, and percentile in the legends of figures 3j; EV 2d.

- Please note that information related to n is missing in the legends of figures 3j; EV 1e; EV 2d; EV 3e.

- Please note that the error bars are not defined in the legends of figures 2d; 3b; EV 3e.

Response: We have now indicated the statistical tests used for data analysis in the legends of Figures 1a, 3h, and 4b. Additionally, the box plots for Figures 3j and EV2d have been clearly defined, including minima, maxima, center, bounds of the box and whiskers, and percentiles.

Information regarding the number of replicates ("n") has been added to the legends of Figures 3j, EV1e, EV2d, and EV3e. In the revised manuscript, we have also defined the error bars for Figures 2d, 3b, and EV3e.

- Please add the primer information provided in the Methods section to the Reagents and tools table and add a callout (e.g. 'see reagents and tools table'). Please also remove the instructions and the empty template table from the reagents and tools table.

Response: We have added primer information to the Reagents and Tools table, along with a callout ("see Reagents and Tools table") in the Methods section. The template instructions and empty table have been removed.

- Please make sure that all the funding information is also entered into the online submission system and that it is complete and similar to the one in the acknowledgement section of the manuscript text file. It seems that the Cytometry and Cell Sorting Core at Baylor College of Medicine with funding from the CPRIT Core Facility Support Award (CPRIT-RP180672) and the NIH grant (CA125123 and RR024574) are missing from the submission system. Please check.

Response: Funding details, including those for the Cytometry and Cell Sorting Core at Baylor College of Medicine, have been added to the online submission system to match the manuscript text.

- Thank you for providing the requested source data. Please upload this as one folder per figure (with all files for one figure in one folder and ZIPed together).

Response: We have organized the source data into folders for each main figure, compressed them into ZIP files, and uploaded them as requested.

In addition, I would need from you uploaded separately:

Response: We have uploaded a two-sentence summary of the manuscript, provided four bullet points highlighting the study's key findings, and included a schematic summary figure illustrating the major findings as a separate file in TIFF format.

Reviewer #1:

The authors have addressed my concerns appropriately. And I have no more comments.

Response: We appreciate the reviewer's positive feedback.

Reviewer #2:

The authors addressed in a satisfactory way all comments raised by reviewers. The authors generated an extensive set of new data and made their conclusions more solid, thus I fully support publication of the revised manuscript.

Response: We appreciate the reviewer's support for the publication of our study.

Reviewer #3:

In this revision, authors sufficiently responded to all my concerns. The revised version shows the difference of double positive cells between 4CL and primed culture much clearer. I found the manuscript is suitable for the publication at EMBO reports.

Response: We appreciate the reviewer's support for the publication of our study.

Please check Fig. 3I before the publication. This figure highlighted differentially expressed genes. I believed the listed genes here are the part of them which are enriched in double positive fraction, but not all the genes. The bottom line is not properly placed, and it might be better show examples of GFP+ genes to clarify.

Response: We apologize for the oversight. We have adjusted the bottom line in the heatmap to accurately represent the enrichment of the highlighted genes in the double-positive fraction. Regarding examples from the GFP+ population, there are only a few genes in this fraction, none of which are particularly relevant to stem cell function or aligned with the main narrative of the study. As such, we have opted to focus on genes enriched in the double-positive population. However, if you would like us to include examples from the GFP+ population, we would be happy to do so. Please let us know.

Dr. Bruno Di Stefano
Baylor College of Medicine
Molecular and Cellular Biology
1 Baylor Plaza
Houston, TX 77030
United States

Dear Dr. Di Stefano,

I am very pleased to accept your manuscript for publication in the next available issue of EMBO reports. Thank you for your contribution to our journal.

Yours sincerely,
